# Factors influencing appropriate use of interventions for management of women experiencing preterm birth: A mixed-methods systematic review and narrative synthesis

**Rana Islamiah Zahroh**[1], **Alya Hazfiarini**[1], **Katherine E. Eddy**[2], **Joshua P. Vogel**[2], **Özge Tunçalp**[3], **Nicole Minckas**[4], **Fernando Althabe**[3], **Olufemi T. Oladapo**[3], **Meghan A. Bohren**[1]*

**1** Gender and Women's Health, Centre for Health Equity, School of Population and Global Health, The University of Melbourne, Carlton, Australia, **2** Maternal, Child and Adolescent Health Program, Burnet Institute, Melbourne, Australia, **3** UNDP/UNFPA/UNICEF/WHO/World Bank Special Programme of Research, Development and Research Training in Human Reproduction (HRP), Department of Sexual and Reproductive Health and Research World Health Organization, Geneva, Switzerland, **4** Department of Maternal, Newborn, Child and Adolescent Health, World Health Organization, Geneva, Switzerland

* meghan.bohren@unimelb.edu.au

## Abstract

### Background

Preterm birth-related complications are the leading cause of death in newborns and children under 5. Health outcomes of preterm newborns can be improved with appropriate use of antenatal corticosteroids (ACSs) to promote fetal lung maturity, tocolytics to delay birth, magnesium sulphate for fetal neuroprotection, and antibiotics for preterm prelabour rupture of membranes. However, there are wide disparities in the rate and consistency in the use of these interventions across settings, which may underlie the differential health outcomes among preterm newborns. We aimed to assess factors (barriers and facilitators) affecting the appropriate use of ACS, tocolytics, magnesium sulphate, and antibiotics to improve preterm birth management.

### Methods and findings

We conducted a mixed-methods systematic review including primary qualitative, quantitative, and mixed-methods studies. We searched MEDLINE, EMBASE, CINAHL, Global Health, and grey literature from inception to 16 May 2022. Eligible studies explored perspectives of women, partners, or community members who experienced preterm birth or were at risk of preterm birth and/or received any of the 4 interventions, health workers providing maternity and newborn care, and other stakeholders involved in maternal care (e.g., facility managers, policymakers). We used an iterative narrative synthesis approach to analysis, assessed methodological limitations using the Mixed Methods Appraisal Tool, and assessed confidence in each qualitative review finding using the GRADE-CERQual approach. Behaviour change models (Theoretical Domains Framework; Capability,

**Data Availability Statement:** All relevant data are within the manuscript and its Supporting Information files.

**Funding:** This research was made possible by the support of the Bill and Melinda Gates Foundation (Grant number: INV-005390) (OTO, JPV) and the UNDP/UNFPA/UNICEF/WHO/World Bank Special Programme of Research, Development and Research Training in Human Reproduction (HRP), a co-sponsored programme executed by the World Health Organization (WHO), and the WHO Department of Maternal, Newborn, Child, Adolescent Health and Ageing (MAB). MAB's time is supported by an Australian Research Council Discovery Early Career Researcher Award (DE200100264) and a Dame Kate Campbell Fellowship (University of Melbourne Faculty of Medicine, Dentistry and Health Sciences). The funders had no role in the study design, data collection and analysis, decision to publish, or preparation of the manuscript.

**Competing interests:** The authors have declared that no competing interests exist.

**Abbreviations:** ACS, antenatal corticosteroid; BEmONC, basic emergency obstetric and newborn care; CEmONC, comprehensive emergency obstetric and newborn care; COM-B, Capability, Opportunity, and Motivation Behaviour model; GBS, Group B Streptococcal Disease; LMIC, low- and middle-income country; MMAT, Mixed Methods Appraisal Tool; PPROM, preterm prelabour rupture of membranes; TDF, Theoretical Domains Framework; WHO, World Health Organisation.

Opportunity, and Motivation (COM-B)) were used to map barriers and facilitators affecting appropriate use of these interventions. We included 46 studies from 32 countries, describing factors affecting use of ACS (32/46 studies), tocolytics (13/46 studies), magnesium sulphate (9/46 studies), and antibiotics (5/46 studies). We identified a range of barriers influencing appropriate use of the 4 interventions globally, which include the following: inaccurate gestational age assessment, inconsistent guidelines, varied knowledge, perceived risks and benefits, perceived uncertainties and constraints in administration, confusion around prescribing and administering authority, and inadequate stock, human resources, and labour and newborn care. Women reported hesitancy in accepting interventions, as they typically learned about them during emergencies. Most included studies were from high-income countries (37/46 studies), which may affect the transferability of these findings to low- or middle-income settings.

## Conclusions

In this study, we identified critical factors affecting implementation of 4 interventions to improve preterm birth management globally. Policymakers and implementers can consider these barriers and facilitators when formulating policies and planning implementation or scale-up of these interventions. Study findings can inform clinical preterm birth guidelines and implementation to ensure that barriers are addressed, and enablers are reinforced to ensure these interventions are widely available and appropriately used globally.

## Author summary

### Why was this study done?

- Complications from preterm birth are the leading cause of death among newborns and children under age 5.

- There are 4 interventions (antenatal corticosteroids, magnesium sulphate, tocolytics, and antibiotics) that can improve health outcomes for preterm newborns, but these interventions are not used correctly or consistently across settings.

- In our research, we explored how and why these 4 interventions are used or not used, in order to help other healthcare providers and families better use them in the future.

### What did the researchers do and find?

- We conducted a systematic review, which means we collected and analysed all relevant research studies about what factors (such as barriers or facilitators) might influence whether or not these 4 interventions are used.

- We found 46 studies, mostly from high-income countries (37 studies), and from the perspectives of women and/or their families (5 studies), healthcare providers (38 studies), or both women and healthcare providers (3 studies).

- We identified several barriers to appropriate use of the 4 interventions, starting with challenges around accurately assessing gestational age, inconsistent clinical guidelines and protocols, healthcare providers' variable knowledge of intervention benefits and harms, and system-level challenges around stock-outs of medicine, limited human resources, and substandard labour and newborn care.

## What do these findings mean?

- Most preterm birth–related deaths happen in low- or middle-income countries (LMICs), but most of the studies we found were from high-income countries, which means that we need to be cautious in applying these findings to LMICs.

- Policymakers and researchers can use these findings when developing policies and planning for scaling up of these interventions, in order to ensure equitable distribution and appropriate use of the interventions globally.

## Introduction

Preterm birth, defined as a birth before 37 weeks gestational age [1], is the leading cause of neonatal mortality worldwide [2]. Nearly 15 million babies are born prematurely every year, accounting for 10.6% of live births worldwide [2]. Importantly, more than 80% of preterm births occur in low- and middle-income countries (LMICs) [2]. There are 4 critical interventions for management of women at risk of preterm birth: antenatal corticosteroids (ACSs), tocolytics, magnesium sulphate, and antibiotics. ACS is the cornerstone intervention, effective in improving preterm birth outcomes by accelerating fetal lung maturation [3–6]. A Cochrane review concluded that when women who are at risk of preterm birth prior to 34 weeks gestation receive ACS, there is a significant reduction in risk of perinatal death, neonatal death, and respiratory distress syndrome, as well as reductions in risk of necrotising enterocolitis, intraventricular haemorrhage, and childhood developmental delays [7]. In addition, tocolytics were historically used to delay the time of birth in the hope of improving preterm birth outcomes. Studies have reported that several tocolytic agents (e.g., betamimetics and calcium channel blockers) reduced imminent preterm birth within 48 hours and 7 days of starting treatment [8,9]. However, uncertainties remain about the benefits of tocolytics, especially in terms of reducing perinatal mortality. Furthermore, magnesium sulphate can be administered to women at risk of early preterm birth for fetal neuroprotection. A Cochrane review found that the risk of babies having gross motor dysfunction and cerebral palsy are significantly reduced in women who received magnesium sulphate [10]. Lastly, antibiotic administration in women with preterm prelabour rupture of membranes (PPROM) is associated with significant reduction in maternal infection [11]. The benefits are also observed in newborns, who have reduced risks of infection, cerebral abnormality, and fewer days in special care [11]. While there are other primary interventions (e.g., smoking cessation programmes) and secondary interventions (e.g., cervical cerclage, progestational agents) for preterm birth, the 2015 World Health Organisation (WHO) recommendations on interventions to improve preterm birth outcomes specifies that the most beneficial set of maternal interventions are those aiming to improve

outcomes for preterm babies when preterm birth is inevitable (e.g., ACS, magnesium sulphate, antibiotics) [12].

Due to these perinatal advantages, many international guidelines recommend ACS administration to women at risk of imminent preterm birth between 24 to 34 weeks gestational age [12–17], magnesium sulphate administration to women between 24 to 35 weeks gestational age [12,18,19], and antibiotics use for women with PPROM [12,20,21]. Tocolytics are generally not recommended for women with imminent risk of preterm birth for the purpose of improving outcomes, however may be used to facilitate ACS administration coverage or referral if needed [12,14,17]. Even though the potential benefits of these interventions to improve outcomes for preterm infants is well recognised, their use at scale varies widely across contexts and settings. These 4 interventions are highly specialised interventions that require certain diagnostic and treatment criteria for eligible women, and specific enabling environments to achieve the desired benefits and minimise harms. Identifying the necessary factors to safely deliver these interventions is critical to achieve effective scale-up for maximal impact at the country level. Previous research has documented potential facilitators and barriers to the use of ACS, tocolytics, and magnesium sulphate [22–24]. However, a critical gap is to understand how these barriers and facilitators can be used in promoting appropriate use and safe scale-up of these 4 interventions globally.

To address this gap, we conducted a global mixed-methods systematic review of factors affecting appropriate use of ACS, magnesium sulphate, tocolytics, and antibiotics for PPROM to improve preterm birth outcomes. The specific objectives are to (1) explore perceptions, preferences, and experiences of women, partners, health providers, and other relevant stakeholders on the use of 4 interventions for preterm birth management; (2) explore how health workers identify women at risk of preterm birth, including assessment of gestational age, identifying signs of maternal infection, and recognising risk of preterm birth; (3) identify factors affecting administration and duration of exposure of the 4 interventions; (4) explore whether the factors affecting appropriate use differ across types of health facilities; and (5) use Theoretical Domains Framework (TDF) and Capability, Opportunity, and Motivation (COM-B) models of behaviour change [25,26] to explore potential strategies in improving appropriate use and scale-up of the 4 interventions.

## Methods

This study is reported as per the Preferred Reporting Items for Systematic Reviews and Meta-Analyses (PRISMA) guideline (S1 Appendix) [27], Enhancing transparency in reporting the synthesis of qualitative research: (ENTREQ) statement (S2 Appendix) [28], and based on guidance from the Cochrane Effective Practice and Organisation of Care group [29]. The review protocol has been published (PROSPERO: CRD42021234509).

### Type of studies

We included primary qualitative, quantitative, and mixed-methods studies addressing or discussing use of ACS, tocolytics, magnesium sulphate, and antibiotics or programme implementation to manage preterm birth. Eligible qualitative studies were those that used qualitative methods for both data collection (e.g., in-depth interviews, focus group discussions, observations) and analysis (e.g., thematic analysis, grounded theory). Eligible quantitative or mixed-methods studies were those that used cross-sectional or mixed-methods approaches for data collection (e.g., surveys, audits). Studies were excluded if they were effectiveness or prevalence studies, or only described guideline formulation processes (without exploring factors affecting use or implementation). Case reports, letters, editorials, commentaries, reviews, study

protocols, posters, and conference abstracts were excluded. There were no limitations on publication date, language, country, or level of healthcare.

## Topic of interest

We included studies where the primary focus was factors affecting use of ACS, tocolytics, magnesium sulphate, and antibiotics (e.g., barriers and facilitators), such as settings for administration, ensuring the right women receive the interventions, and duration of exposure (Fig 1). We assessed the identified factors of use against appropriate use, defined as adhering to WHO recommendations for preterm birth interventions; see Table 1 [12,13]. Eligible studies explored perspectives of women, partners or community members who experienced preterm birth or were at risk of preterm birth and/or received any of the 4 interventions, health workers providing maternity and newborn care (e.g., midwives, nurses, doctors), and other stakeholders involved indirectly in maternal care (e.g., facility managers, policymakers).

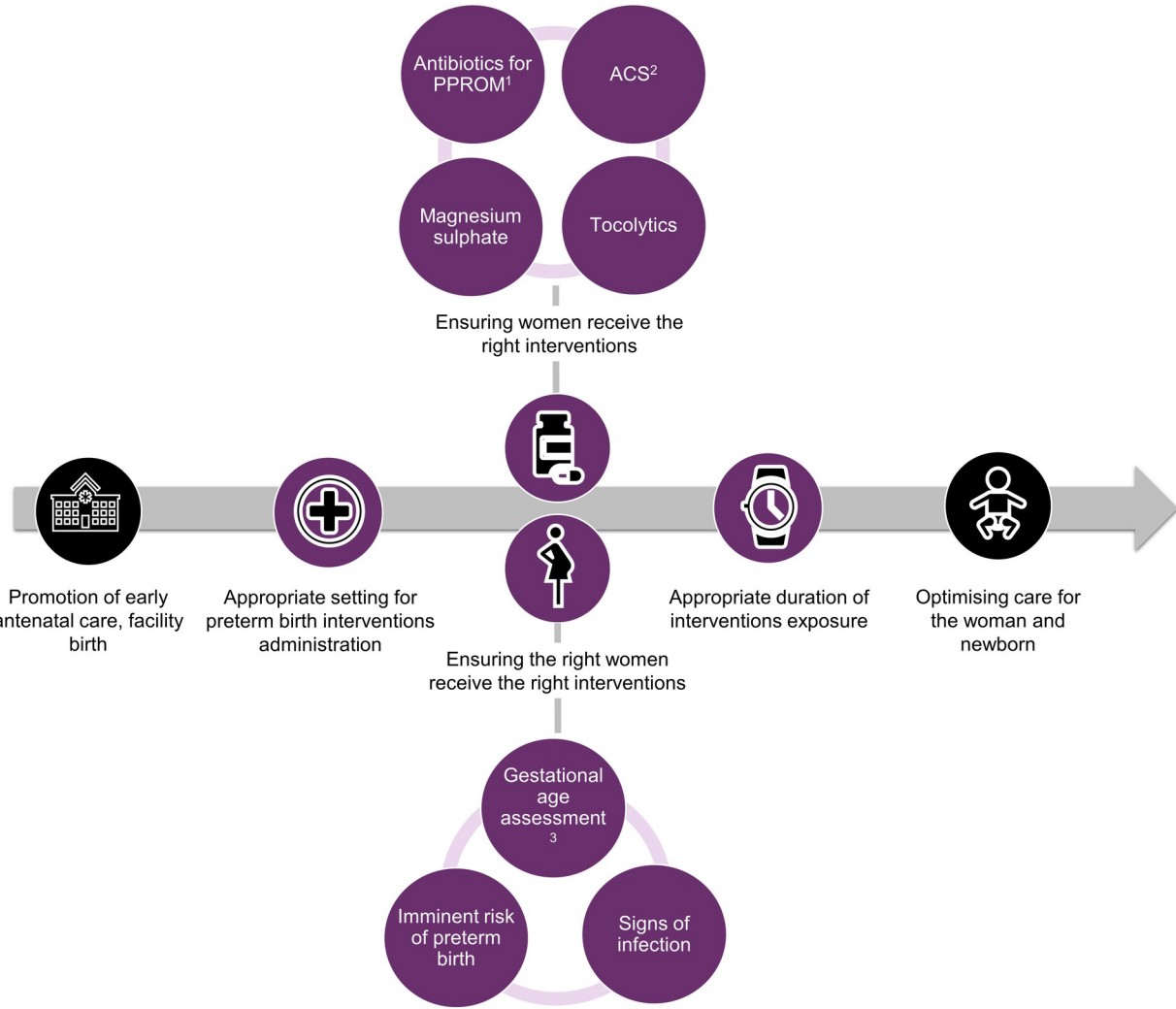

**Fig 1. Scope of this review, where purple colour depicts inclusion in the review.** [1]Preterm Premature Rupture of Membrane (PPROM). [2]Antenatal corticosteroids (ACS). [3]At antenatal care and/or point of care.

**Table 1. Definition of appropriate use of interventions, based on WHO recommendations\*.**

| Domain | ACS | Tocolytics | Magnesium sulphate | Antibiotics |
|---|---|---|---|---|
| Who | Women at risk of imminent preterm birth (birth is predicted to occur within 7 days starting treatment) with no clinical evidence of infection | Women at risk of imminent preterm birth who are eligible for ACSs administration | Women at risk of imminent preterm birth | Women with PPROM |
| When | Gestational age from 24 to 34 weeks accurately assessed through ultrasound dating | N/A | Gestational age less than 32 weeks assessed by ultrasound dating | After a definitive diagnosis of PPROM |
| Where | Health facility where adequate childbirth and preterm newborn care are available (including resuscitation, thermal care, feeding support, infection treatment, and safe oxygen use) | Health facility where adequate childbirth is available | Health facility where adequate childbirth is available | Health facility where adequate childbirth is available |
| How | Intramuscular dexamethasone or betamethasone (24 mg in divided doses). Single repeat course can be administered if birth does not occur within 7 days of initial dose and there is high risk of preterm birth in the next 7 days | Nifedipine (a calcium channel blocker) is the preferred agent, administered as 10–30 mg initial dose, followed by 10–20 mg every 4–8 hours up to 48 hours or until referral complete | Administer prior to birth or up to 24 hours prior to anticipated birth | Erythromycin-recommended regimen |

\*Adapted from WHO recommendations on interventions to improve preterm birth outcomes: evidence base [12]; N/A, not applicable.

ACS, antenatal corticosteroid; PPROM, preterm prelabour rupture of membranes; WHO, World Health Organisation.

## Search methods for identification of studies

We searched MEDLINE, EMBASE, CINAHL, and Global Health databases from the inception date to 16 May 2022. Search strategies were developed in consultation with an information specialist and used combinations of terms related to preterm birth, ACS, tocolytics, magnesium sulphate for fetal neuroprotection, and antibiotics for PPROM (S3 Appendix). We searched grey literature using Open Grey Literature and Google search, where we examined the first 10 pages of the results for each intervention.

## Selection of studies

We imported the search results into Covidence (Covidence systematic review software, Veritas Health Innovation, Melbourne, Australia), and at least 3 reviewers (RIZ, KEE, and MAB) independently reviewed title and abstracts to evaluate eligibility against the prespecified criteria. Google Translate was used to translate titles and abstracts published in languages other than those the review team are proficient in (English, Bahasa Indonesia, Korean, French, Spanish, and Turkish). We retrieved the full text of all papers identified as potentially relevant by one or both reviewers, and 2 reviewers assessed eligibility independently (RIZ and KEE), with disagreements resolved through discussion with 2 reviewers (MAB and JPV). If the translated title and abstracts were potentially relevant for inclusion, the full text was translated first using Google Translate, and then translation was checked and corrected by a native speaker if full text inclusion was indicated.

## Data extraction and assessing methodological limitations

Two reviewers (RIZ and KEE) extracted relevant data using a form designed for this review, including the following information: study settings, participant characteristics, objectives, any framework used, methodology, study design, recruitment, data collection and analysis methods, findings, and conclusions. Themes, interpretation, and quotations were extracted from qualitative studies, while numbers and interpretations were extracted from quantitative findings. One reviewer extracted relevant data, which was then double checked by the second reviewer. The data extraction form was pre-tested on three eligible studies and refined.

Three reviewers (RIZ, KEE, and AH) assessed methodological limitations for each study using an adapted Mixed Methods Appraisal Tool (MMAT) (use of critical appraisal tool changed from protocol version) [30]. For qualitative studies, we assessed study aims, methodology selection, design, recruitment, data collection, data analysis, coherence, reflexivity, and ethical considerations. For quantitative studies, we assessed sampling strategy, sample representativeness, appropriateness of measurement tools, response rates, selective reporting, statistical analysis, and other potential sources of bias and confounding. For mixed-methods studies, we assessed rationale, appropriateness in addressing research questions, integration of results, explanation of inconsistencies, and adherence to each methodological stream. Any disagreement was resolved through discussion, and when required, by involving a third reviewer (MAB). The quality rating was not used to exclude any studies and instead use to assess confidence in the evidence. We report the methodological limitations assessments in S4 Appendix.

## Data management, analysis, and synthesis

We used an iterative narrative synthesis approach to analysis [31], by developing a synthesis of findings of included studies, exploring relationships in the data, and assessing robustness of the synthesis. First, we conducted an inductive thematic synthesis of qualitative data [32]. This step included line-by-line coding of findings from 6 included qualitative studies with thick data; based on this preliminary coding, we developed a qualitative codebook. We used this codebook to code the remaining qualitative studies and organised the codes into a hierarchy based on relationships between emerging concepts using NVivo 12 (NVivo, Melbourne, Australia: QSR International; Version 12 for Windows). Two reviewers (MAB and RIZ) used the coded qualitative data to develop qualitative review findings by iteratively exploring and discussing emergent themes and concepts.

Next, we mapped quantitative data to the qualitative review findings, to explore areas of convergence and divergence, or where the quantitative evidence extended our understanding of the qualitative evidence. Finally, we mapped both qualitative and quantitative findings to the TDF and COM-B models of behaviour change [25,26] to clarify how identified barriers and facilitators may influence individual and collective behaviours. TDF and COM-B are interrelated behaviour change models, where each of the 14 TDF domains (knowledge, skills, social and professional role and identity, beliefs about capabilities, optimism, reinforcement, intentions, goals, memory, attention and decision processes, environmental context and resources, social influences, emotion, and behavioural regulation) maps uniquely to the COM-B components (capability, opportunity, and motivation). COM-B is a comprehensive behaviour change model that provides a framework to assess 3 fundamental conditions that must be understood and addressed to promote behaviour change. We defined 2 behaviours for the purposes of the mapping, based on the scope of the review questions (Fig 1) and previously known threats to implementation: (1) appropriate use of the 4 interventions by providers; and (2) acceptability of the 4 interventions by women. We firstly mapped facilitators and barriers to the 14 domains of the TDF, then mapped to the 3 COM-B domains [33]. For example, healthcare provider "lack of awareness on ultrasound dating for gestational age" mapped to the Capability-Knowledge domain. After mapping, we then identified potential strategies from each of 3 COM-B domains to promote the 2 intended behaviours [26].

We assessed confidence in qualitative review findings using the GRADE-CERQual approach [34,35] and considered respective critical appraisal results for quantitative review findings. Three review authors (RIZ, AH, and MAB) conducted GRADE-CERQual assessments based on 4 components: methodological limitations [36], coherence [37], adequacy of data [38], and relevance [39]. Each component was assessed by the level of concerns (no or

very minor, minor, moderate, and serious) [34–39]. Then, we made a judgement about the overall confidence in review finding (high, moderate, low, or very low) [34–39]. All findings started with high confidence and were downgraded if there were important concerns regarding any components. We present the summaries of qualitative findings and GRADE-CERQual assessments in Table 2 and the full evidence profile in S5 Appendix. Summarised quantitative findings are included in S6 Appendix.

## Results

We identified 15,878 citations from database searches, 13 citations from grey literature, and included 46 studies (Fig 2. PRISMA flowchart). These studies were published between 1987 and 16 May 2022 and reported in English, Spanish, and Mandarin.

Table 2 reports the characteristics of included studies. In summary, the 46 included studies were conducted in 32 countries in Region of the Americas (6 countries: United States of America [24,40–51], Canada [44,52–54], Mexico [55,56], Ecuador [56], El Salvador [56], and Uruguay [56]), Western Pacific Region (7 countries: Australia [23,57–63],New Zealand [58–64], Vietnam [22], Singapore [65], Taiwan [66,67], Cambodia [68], and Philippines [68]), European Region (4 countries: United Kingdom [69–72], Ireland [73], Sweden [74], and France [75,76]), Southeast Asia Region (4 countries: Thailand [77], Bangladesh [22], India [22,78], and Nepal [22]), Eastern Mediterranean Region (2 countries: Afghanistan [22] and Pakistan [22]), Africa region (9 countries: Cameroon [22], Democratic Republic of Congo [22,79], Kenya [22], Malawi [22,79–81], Nigeria [22,79], Uganda [22,79], Ethiopia [79], Sierra Leone [79], and Tanzania [79]), including 3 multiregion studies [22,56,79]. Most studies were conducted in high-income countries (37/46 studies) [23,24,40–54,57–67,69–76,82], with 9 studies conducted in LMICs [22,55,56,68,77–81].

Five studies included perspectives of women and/or their partners [51,52,62,63,67], 3 studies included both women's and provider's perspectives [78,80,81], and the remaining 38 studies included only health providers' perspectives [22–24,40–50,53–61,64–66,68–77,79,82].

Thirty-two studies used quantitative methods (typically surveys) [22,41–50,52–61,64,69–71,73–78], 11 studies used qualitative methods (typically in-depth interviews or focus group discussions) [23,24,40,51,62,63,66,67,79–81], and 3 studies used mixed-methods (audit and feedback, qualitative evaluation) [68,72,82].

Most studies (32/46) reported factors on ACS use [22,24,40–47,49,50,52,53,55–57,59,61–64,68–70,73,74,77–81], while comparatively fewer reported on tocolytics (13/46) [24,40,44,51,53,54,57,59,66,67,73,76,80], magnesium sulphate for fetal neuroprotection (9/46) [23,42,58,60,65,72,75,80,82], or antibiotics for PPROM use (5/46) [48,57,70,71,73], with some studies reported use on more than one intervention, with mostly reported ACS and tocolytics at the same time [24,40,42,44,53,57,59,70,73,80] (S7 Appendix).

Detailed critical appraisals of included studies are available in S4 Appendix. For qualitative studies, many studies reported insufficiently detailed and unjustified recruitment strategies, limited elaboration on data analysis methods, minimal interpretation and use of quotations, missing details on ethical considerations, and importantly many studies did not include a reflexivity statement. Across quantitative studies, the primary concerns were regarding the appropriateness of measurement tools, sample representativeness, unclarity on risk of nonresponse bias, and statistical analysis was not elaborated. In mixed-methods studies, the rationale for using the methodology, integration, and interpretation of the qualitative and quantitative data were often missing. All included studies were published in peer-reviewed journals, except one dissertation [51].

**Table 2. Characteristics of included studies.**

| Author | Title | Country | ACS | Tocolytics | MgSO4 | Antibiotics | Designs | Sample size | Participants |
|---|---|---|---|---|---|---|---|---|---|
| Tucker Edmonds 2015 [41] | The influence of resuscitation preferences on obstetrical management of periviable deliveries | United States of America | ☑ | | | | Survey | 295 | Obstetrician gynaecologists |
| Vargas-Origel 2000 [55] | ACS. Its use and the obstetrician attitudes. | Mexico | ☑ | ☑ | | | Prospective observational with survey | 48 | Obstetricians |
| Buchanan 2004 [57] | Preterm prelabour rupture of the membranes: a survey of current practice | Australia | ☑ | | | ☑ | Questionnaire | 731 | Obstetricians |
| Bousleiman 2015 [42] | Use and attitudes of obstetricians toward 3 high-risk interventions in MFMU Network hospitals | United States of America | ☑ | | ☑ | | Survey | 329 | Obstetricians |
| Battarbee 2020 [43] | Management of diabetic women with threatened preterm birth: a survey of Maternal-Fetal Medicine providers | United States of America | ☑ | | | | Survey | 159 | MFM providers |
| Chan 2006 [69] | Staff views on the management of the extremely preterm infant | United Kingdom | ☑ | | | | Questionnaire | 69 | Obstetricians, neonatologists, midwives, and neonatal nurses |
| Capeless 1987 [44] | Management of preterm premature rupture of membranes: lack of a national consensus | United States of America and Canada | ☑ | ☑ | | | Questionnaire | 285 | Obstetricians |
| Bain 2013 [58] | Implementation of a clinical practice guideline for antenatal magnesium sulphate for neuroprotection in Australia and New Zealand | Australia and New Zealand | | | ☑ | | Survey | 25 | Obstetricians, trainee medical officers, midwives |
| Aghajafari 2002 [52] | Multiple versus single courses of ACS for preterm birth: a pilot study | Canada | ☑ | | | | Randomised controlled trial with questionnaire | 12 | Women |
| Hueston 1997 [45] | Variations between family physicians and obstetricians in the evaluation and treatment of preterm labour | United States of America | ☑ | | | | Questionnaire | 321 | Obstetrician gynaecologists and family physicians |
| Smith 2011 [73] | Practices for predicting and preventing preterm birth in Ireland: a national survey | Ireland | ☑ | ☑ | | ☑ | Questionnaire | 66 | Obstetrician gynaecologists |
| Battarbee 2019 [46] | Practice Variation in Antenatal Steroid Administration for Anticipated Late Preterm Birth: A Physician Survey | United States of America | ☑ | | | ☑ | Survey | 193 | Obstetricians |

*(Continued)*

**Table 2.** (Continued)

| Author | Title | Country | ACS | Tocolytics | MgSO4 | Antibiotics | Designs | Sample size | Participants |
|--------|-------|---------|-----|-----------|-------|------------|---------|------------|-------------|
| Danerek 2012 [74] | Attitudes of Swedish midwives towards management of extremely preterm labour and birth | Sweden | ☑ | | | | Questionnaire | 259 | Midwives |
| Hutton 1989 [64] | New Zealand obstetricians' management of hypertension in pregnancy. A questionnaire survey | New Zealand | ☑ | | | | Questionnaire | 65 | Obstetricians |
| Erickson 2001 [47] | Obstetrician-gynaecologists' knowledge and training about ACS | United States of America | ☑ | | | | Questionnaire | 487 | Obstetricians |
| Cook 2004 [59] | Survey of the management of preterm labour in Australia and New Zealand in 2002 | Australia and New Zealand | ☑ | ☑ | | | Questionnaire | 813 | Obstetrician gynaecologists |
| Gatman 2020 [60] | Survey on use of antenatal magnesium sulphate for fetal neuroprotection prior to preterm birth in Australia and New Zealand: Ongoing barriers and enablers | Australia and New Zealand | | | ☑ | | Questionnaire | 24 | Obstetrician, midwives, neonatologists |
| Glass 2005 [48] | Opportunities to reduce overuse of antibiotics for perinatal group B streptococcal disease prevention and management of preterm premature rupture of membranes | United States of America | | | | ☑ | Questionnaire | 519 | Obstetrician gynaecologists |
| Chollat 2017 [75] | Antenatal magnesium sulphate administration for fetal neuroprotection: a French national survey | France | | | ☑ | | Online and phone survey | 138 | Obstetricians, anaesthetists, neonatologists |
| Aleman 2013 [56] | Use of ACS for preterm birth in Latin America: providers knowledge, attitudes, and practices | Ecuador, El Salvador, Mexico and Uruguay | ☑ | | | | Questionnaire | 353 | Physicians, midwives, nurse, in some cases medical students |
| Baker 2015 [53] | Current practices in the prediction and prevention of preterm birth in patients with higher-order multiple gestations | Canada | ☑ | ☑ | | | Survey | 81 | MFM specialists |
| Saengwaree 2005 [77] | Changing physician's practice on ACS in preterm birth | Thailand | ☑ | | | | Medical records review and questionnaire | 39 (questionnaire only) | Obstetricians |
| Wilson 2002 [70] | The Leeds University Maternity Audit Project | United Kingdom | ☑ | | | ☑ | Case note audit and interview | 88 taped interviews | Obstetricians, unit manager, paediatrician, midwife |

(*Continued*)

**Table 2.** (Continued)

| Author | Title | Country | ACS | Tocolytics | MgSO4 | Antibiotics | Designs | Sample size | Participants |
|---|---|---|---|---|---|---|---|---|---|
| Hong 2017 [65] | Resident Quality Improvement Project: Antenatal Magnesium Sulfate Protocol for Fetal Neuroprotection in Preterm Births | Singapore | | | ☑ | ☑ | Prospective study with audit and survey | 42 | Staff members of the hospital |
| Hui 2007 [54] | Preterm Labour and Birth: A Survey of Clinical Practice Regarding Use of Tocolytics, ACS, and Progesterone | Canada | | ☑ | | | Survey | 2,821 | Obstetricians |
| Kenyon 2010 [71] | Has publication of the results of the ORACLE Children Study changed practice in the UK? | United Kingdom | | | | ☑ | Questionnaire | 324 | Obstetricians |
| McGoldrick 2017 [61] | Investigating antenatal corticosteroid clinical guideline practice at an organisational level | Australia and New Zealand | ☑ | | | | Questionnaire | 40 | Clinical managers at 27 secondary and 25 tertiary maternity hospitals |
| TuckerEdmonds 2015 [50] | A national survey of obstetricians' attitudes toward and practice of periviable intervention | United States of America | ☑ | | | | Questionnaire | 310 | Obstetrician gynaecologists, MFM specialists |
| Rousseau 2020 [76] | Do obstetricians apply the national guidelines? A vignette-based study assessing practices for the prevention of preterm birth | France | | ☑ | | | Survey (structured and open-ended questions) | 423 | Obstetricians |
| Tucker Edmonds 2015 [49] | Comparing obstetricians' and neonatologists' approaches to periviable counselling | United States of America | ☑ | | | | Exploratory simulation study | 31 | Obstetricians and neonatologists |
| Liu 2015 [22] | ACS for management of preterm birth: a multi-country analysis of health system bottlenecks and potential solutions | Afghanistan, Cameroon, Democratic Republic of Congo, Kenya, Malawi, Nigeria, Uganda, Bangladesh, India, Nepal, Pakistan, and Vietnam | ☑ | | | | Maternal-newborn bottleneck analysis through workshop and survey | Not specified | Health providers and policymakers |
| Kankaria 2021 [78] | Readiness to Provide Antenatal Corticosteroids for Threatened Preterm Birth in Public Health Facilities in Northern India | India | ☑ | | | ☑ | Cross-sectional through facility assessment, semistructured questionnaire, report summary | 107 | Health providers and women |
| McGoldrick 2016 [62] | Consumers attitudes and beliefs towards the receipt of ACS and use of clinical practice guidelines | Australia and New Zealand | | ☑ | | | Qualitative interviews and open-ended questionnaire | 24 | Women |

(*Continued*)

**Table 2.** (Continued)

| Author | Title | Country | ACS | Tocolytics | MgSO4 | Antibiotics | Designs | Sample size | Participants |
|---|---|---|---|---|---|---|---|---|---|
| Hsieh 2006 [67] | The lived experience of first-time expectant fathers whose spouses are tocolyzed in hospital | Taiwan | | ☑ | | | Qualitative interviews | 6 | Partners of women |
| Greensides 2018 [79] | ACS for women at risk of imminent preterm birth in 7 sub-Saharan African countries: a policy and implementation landscape analysis | Democratic Republic of the Congo, Ethiopia, Malawi, Nigeria, Sierra Leone, Tanzania, and Uganda | ☑ | | | | Document reviews and qualitative interviews | 12 | Senior-level ministry of health representative, and organisations working closely with Ministry of Health |
| Bain 2015 [23] | Barriers and enablers to implementing antenatal magnesium sulphate for fetal neuroprotection guidelines: a study using the theoretical domains framework | Australia | | | | | Qualitative interviews | 45 | Obstetricians, midwives, neonatologists |
| Antony 2019 [80] | Qualitative assessment of knowledge transfer regarding preterm birth in Malawi following the implementation of targeted health messages over 3 years | Malawi | ☑ | | | | Focus group discussions | 70 | Nurse midwives, CHWs, nurses, matrons, clinic land medical officers, medical and dental assistants, health surveillance assistants |
| Kaplan 2016 [40] | Reliable implementation of evidence: a qualitative study of antenatal corticosteroid administration in Ohio hospitals | United States of America | ☑ | ☑ | | | Focus group discussions, qualitative interviews | 97 | Obstetricians, physician trainees, nurse midwives, nurses |
| Levison 2014 [81] | Qualitative assessment of attitudes and knowledge on preterm birth in Malawi and within country framework of care | Malawi | ☑ | | | | Focus group discussions, incidence narrative, qualitative interviews | 33 participants on focus groups, unclear how many were interviewed | Women, partners, community health workers, nurse midwife/matrons, clinical officers (physician) |
| Leviton 1995 [24] | An exploration of opinion and practice patterns affecting low use of ACS | United States of America | ☑ | ☑ | | | Qualitative interviews and focus group discussions | 8 interview participants; 4 focus groups (total not stated but 8–15 participants on each group) | Obstetricians and neonatologists |
| McGoldrick 2016 [63] | Identifying the barriers and enablers in the implementation of the New Zealand and Australian Antenatal Corticosteroid Clinical Practice Guidelines | Australia and New Zealand | ☑ | | | | Qualitative interviews or open-ended questionnaire | 73 | Obstetricians, midwives, neonatologists, paediatricians |
| Hu 2006 [66] | Study of stress and coping behaviours in families of hospitalized pregnant woman undergoing tocolysis | Taiwan | | ☑ | | | Qualitative interviews | Unclear | Women's partners |

(*Continued*)

**Table 2.** (Continued)

| Author | Title | Country | ACS | Tocolytics | MgSO4 | Antibiotics | Designs | Sample size | Participants |
|--------|-------|---------|-----|-----------|-------|-------------|---------|-------------|--------------|
| Kalb 1993 [51] | Women's experiences using terbutaline pump therapy for the management of preterm labour | United States of America | | ☑ | | ☑ | Qualitative interviews | 12 | Women |
| Smith 2016 [68] | Providing ACS for preterm birth: a quality improvement initiative in Cambodia and the Philippines | Cambodia and Philippines | ☑ | | | | Pre- and post-intervention design with monthly audit and feedback sessions (written data) | Not specified | Maternity care staffs that participate in audit process |
| Burhouse 2017 [72] | Preventing cerebral palsy in preterm labour: a multiorganisational quality improvement approach to the adoption and spread of magnesium sulphate for neuroprotection | United Kingdom | | | ☑ | | Quality improvement study with qualitative evaluation (focus groups, surveys, quantitative data capture) | Not specified | Medical staffs: only midwives mentioned |
| Teela 2015 [82] | Magnesium sulphate for fetal neuroprotection: benefits and challenges of a systematic knowledge translation project in Canada | Canada | | | ☑ | | Focus group discussions, site visits, survey | 188 survey respondents | Physicians, nurses, midwives, residents, students, pharmacist, administrators |

CHW, community health worker; MFM, maternal-fetal medicine.

## Results of qualitative and quantitative synthesis

We identified 8 overarching themes in the qualitative evidence synthesis: (1) inaccurate assessment of gestational age; (2) inconsistent practice guidelines; (3) variable knowledge about the interventions; (4) providers' perceived risks and benefits; (5) barriers in administration of interventions; (6) appropriate settings for administration; (7) strategies to improve appropriate use; and (8) women's perspectives and experiences (S8 Appendix. Development of themes). Within these overarching themes, we developed 27 qualitative findings (Table 3. Summary of qualitative findings) and used the GRADE-CERQual approach to assess confidence. Eight findings were assessed as high confidence, 17 as moderate confidence, and 2 as low confidence. The explanation for each GRADE-CERQual assessment is shown in S5 Appendix. GRADE-CERQual Evidence Profile. The summaries of qualitative findings were mostly similar across interventions and settings; where there were differences, we highlight these below. After developing the summary of qualitative findings, quantitative evidence was descriptively mapped to these findings to explore areas of convergence or divergence (S6 Appendix).

## Inaccurate assessment of gestational age

**Limitations about determining gestational age.** Women and health providers reported that last menstrual period or last month of menstrual period were the most common methods in assessing gestational age in LMICs, despite health providers acknowledging their limited accuracy. Some health providers in these settings were aware of ultrasound assessments of gestational age, whereas community health workers were not aware on the role of ultrasound dating in pregnancy (1.1 –Moderate Confidence) [68,80,81]. Last menstrual period was often not

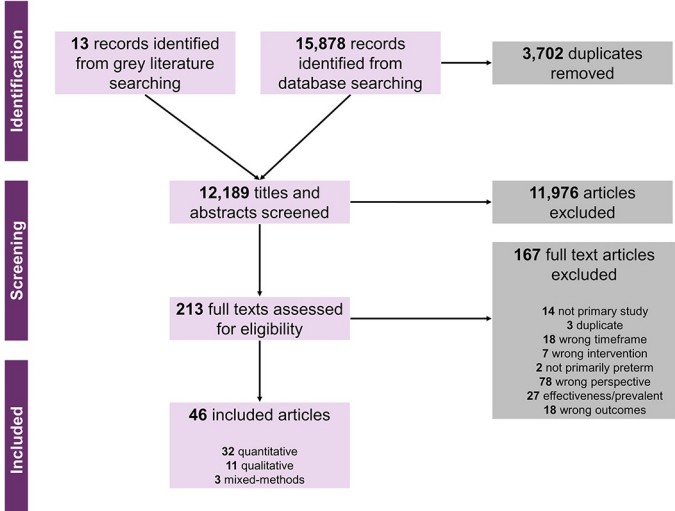

**Fig 2. PRISMA flowchart depicting search and selection process.**

known or not reported by women, making it difficult to assess gestational age [68]. Other methods used included calculating month of missed period, using a gestational wheel, or using first antenatal appointment as proxies for gestational age assessment [68,80,81]. No qualitative studies explored limitations of gestational age assessment using ultrasound.

Quantitative evidence extended the understanding of qualitative evidence that accurate and reliable gestational age assessments in LMICs was limited [78]. Ultrasound gestational age dating was typically only available at higher-level hospitals [78], which may hinder appropriate use of ACS. Similarly, to qualitative evidence, health providers routinely used fundal height, followed by last menstrual period and ultrasound [78].

## Inconsistent practice guidelines

**Inconsistent practice guidelines.** There were substantial variations in the content of practice and implementation guidelines on ACS across contexts, typically about appropriate gestational age criteria, determination of what constitutes imminence in preterm labour birth, how to exclude maternal infection that precludes ACS use, adequacy of childbirth and preterm newborn care environment, and use in specific populations of women (2.1 –Moderate confidence) [63,79]. Despite these variations, health providers placed high value on use of ACS and magnesium sulphate clinical practice guidelines as professional standards and perceived them as a facilitator of use [63,82]. Most health providers expressed the need to improve guidelines on ACS and magnesium sulphate by making them more user-friendly and improving accessibility and dissemination [23,40,63].

Quantitative evidence supported the qualitative findings that the presence and content of guidelines or policy on ACS and magnesium sulphate varies across settings at national and health facility levels [22,56,65,75,78]. Where country-level policy and guidelines for ACS existed, they were perceived as outdated, unclear, or were not widely disseminated [22,56,65,75].

## Variable knowledge about the interventions

**Health providers' knowledge of the interventions.** Health providers' knowledge about guidelines for and use of ACS, magnesium sulphate, and tocolytics was variable. Where there

**Table 3. Summary of qualitative findings.**

| # | Summary of qualitative review findings | Contributing qualitative studies | Overall CERQual assessment | Explanation of overall assessment |
|---|---|---|---|---|
| 1 | **Inaccurate assessment of gestational age** | | | |
| 1.1 | **Limitations about determining gestational age**<br>Women and health providers reported that last menstrual period or last month of menstrual period were the most common methods in assessing gestational age in LMICs, despite health providers acknowledging their limited accuracy. Some health providers in these settings were aware of ultrasound assessments of gestational age, whereas community health workers were not aware on the role of ultrasound dating in pregnancy. | [68,80,81] | **Moderate confidence** | No or very minor concerns on coherence, yet we downgraded due to minor concerns regarding adequacy, and moderate concerns on methodological limitations and relevance. |
| 2 | **Inconsistent practice guidelines** | | | |
| 2.1 | **Inconsistent practice guidelines**<br>There were substantial variations in the content of practice and implementation guidelines on ACS across contexts, typically about appropriate gestational age criteria, determination of what constitutes imminence in preterm labour birth, how to exclude maternal infection that precludes ACS use, adequacy of childbirth and preterm newborn care environment, and use in specific populations of women. | [63,79] | **Moderate confidence** | No or very minor concerns regarding coherence and relevance but downgraded due to minor concerns regarding adequacy and moderate concerns regarding methodological limitations. |
| 3 | **Variable knowledge about the interventions** | | | |
| 3.1 | **Health providers' knowledge of the interventions**<br>Health providers' knowledge about guidelines for and use of ACS, magnesium sulphate, and tocolytics was variable. Where there was high levels of knowledge and experience in administering the interventions, this improved implementation feasibility. Lack of knowledge or outdated knowledge were barriers to appropriate use. The key existing knowledge gaps were related to differences between research evidence and previous clinical training or experience, which sometimes involved different courses, dosing, and duration. | [23,24,40,63,80–82] | **Moderate confidence** | No or very minor concerns regarding coherence and adequacy but downgraded due to moderate concerns regarding methodological limitations and relevance. |
| 3.2 | **Knowledge about optimal gestational age for intervention administration**<br>Knowledge about optimal gestational age for administration of ACS and magnesium sulphate varied across health providers, with mixed opinions about the earliest gestational age they would administer and agreement that these were challenging to have with women and families. Opinion about optimal gestational age for administration of interventions were also balanced with other factors including estimated time to birth, threatened versus imminent preterm birth, and local standards of practice. | [24,63,79,82] | **Moderate confidence** | No or very minor concerns regarding coherence, but we downgraded due to minor concerns regarding adequacy, and moderate concerns regarding relevance as well as serious concerns regarding methodological limitations. |
| 4 | **Perceived risks and benefits** | | | |
| 4.1 | **Uncertainties in prescribing and administering ACS for specific populations of women**<br>Health providers had uncertainties and lacked confidence regarding certain aspects of prescribing and administering ACS, such as whether to use repeat doses, or whether to use ACS in specific clinical situations (such as in women with diabetes, hypertension, fetal complications, maternal infection, or PPROM). | [40,63,80] | **Moderate confidence** | No or very minor concerns regarding coherence yet downgraded due to minor concerns regarding adequacy as well as moderate concerns regarding methodological limitations and relevance. |

*(Continued)*

**Table 3.** (Continued)

| # | Summary of qualitative review findings | Contributing qualitative studies | Overall CERQual assessment | Explanation of overall assessment |
|---|---|---|---|---|
| 1 | **Inaccurate assessment of gestational age** | | | |
| 4.2 | **Scepticism of the evidence base for interventions** Health providers had mixed beliefs about the evidence supporting ACS and magnesium sulphate for fetal neuroprotection. While some providers agreed with and believed in the evidence supporting their use, others were sceptical about long-term outcomes, availability of high-quality trials, mixed evidence of effects and sufficiency of evidence, all of which may act as barriers to use. | [23,24,40,63,82] | **Moderate confidence** | No or very minor concerns regarding coherence but downgraded due to minor concerns regarding adequacy, moderate concerns regarding methodological limitations and relevance. |
| 4.3 | **Beliefs about risks of interventions** While many health providers believed that risks of ACS and magnesium sulphate were negligible, some had concerns about possible safety issues (particularly interactions with tocolytics, exacerbation of pulmonary oedema), low tolerance by women, long-term risks of complications for women, whether use at earlier gestational age is appropriate (<28 weeks), and risk of maternal infection. | [23,24,79,80] | **Moderate confidence** | No or very minor concerns regarding coherence, but downgraded due to minor concerns regarding adequacy, and moderate concerns regarding methodological limitations as well as relevance. |
| 4.4 | **Beliefs about risks of interventions—interaction with tocolytics** Some health providers believed that interaction of magnesium sulphate and ACS individually with tocolytic agents (particularly nifedipine) is associated with exacerbated adverse effects and toxicity for women. This belief may hinder administration of magnesium sulphate and ACS, in women who are also eligible for ACS and tocolytics. | [23,24] | **Low confidence** | No or very minor concerns regarding coherence but downgraded due to moderate concerns regarding methodological limitations and relevance as well as serious concerns regarding adequacy. |
| 4.5 | **Beliefs about benefits of interventions** Most health providers recognised the benefits of magnesium sulphate and ACS, believing that these interventions save lives, and benefits mostly outweigh risks. Women similarly believed that administration of ACS is beneficial, stressing the importance of using only when necessary and receiving information about potential side effects. In contrast, many health providers believed that tocolytics do not work and do not stop labour. | [23,24,40,62,63,80] | **High confidence** | No or very minor concerns regarding coherence and adequacy yet downgraded due to minor concerns regarding methodological limitations and moderate concerns regarding relevance. |
| 5 | **Barriers in administration of interventions** | | | |
| 5.1 | **Uncertainties on when to administer interventions** The unpredictability of preterm birth, including difficulty diagnosing threatened versus imminent preterm birth, can lead to provider hesitation in administering ACS and magnesium sulphate—providers fear being held responsible or blamed for potentially unnecessary treatment. To cope with these uncertainties, providers may delay treatment, preferring a "wait and see" approach. | [23,24,63,80,82] | **Moderate confidence** | No or very minor concerns regarding coherence yet downgraded due to minor concerns regarding adequacy as well as moderate concerns regarding methodological limitations and relevance. |
| 5.2 | **Time constraints and complexity in prescribing and administering** Health providers described time constraints in prescribing and administering ACS and magnesium sulphate as a critical overarching barrier to appropriate use, due to the acute nature and time pressures of imminent preterm birth, high intensity of workload, and competing tasks. Many health providers believed that prescribing and administering magnesium sulphate is complex, as preparation takes too much time, or is difficult to "draw it all up," which could deter health providers in administering the medication when they feel under pressure. | [23,63,82] | **Moderate confidence** | No or very minor concerns regarding coherence, yet downgraded due to moderate concerns regarding methodological limitations, relevance, and adequacy. |

*(Continued)*

**Table 3.** (Continued)

| # | Summary of qualitative review findings | Contributing qualitative studies | Overall CERQual assessment | Explanation of overall assessment |
|---|---|---|---|---|
| 1 | **Inaccurate assessment of gestational age** | | | |
| 5.3 | **Stocking medications in maternity ward**<br>Maintaining consistent stock of ACS and magnesium sulphate that is readily available in the maternity ward and emergency department, and the availability of health providers who are readily able to assess women in preterm labour, was critical to ensure that women received prompt treatment. | [23,40,68,79,80] | **High confidence** | No or very minor concerns regarding coherence, yet downgraded due to minor concerns regarding relevance, adequacy, and moderate concerns regarding methodological limitations. |
| 5.4 | **Regulatory policies and beliefs about prescribing and administering authority**<br>National-level guidance is often limited about who can prescribe and administer ACS and magnesium sulphate; where there is guidance, typically only obstetricians are authorised to prescribe and administer, while other health providers can administer under clinical oversight, but not prescribe. Many health providers (obstetricians, neonatologists, midwives) likewise believe that prescription and administration of ACS and magnesium sulphate should be prescribed and administered by obstetricians-only, even though multidisciplinary decision-making was highly valued. | [23,40,63,79] | **High confidence** | No or very minor concerns regarding coherence and relevance yet downgraded due to minor concerns on methodological limitations and adequacy. |
| 6 | **Appropriate settings for administration** | | | |
| 6.1 | **Appropriate settings for ACS administration**<br>In some national guidelines and in clinical practice, administration of ACS is allowed at only at tertiary facilities where CEmONC and essential preterm newborn care interventions are available. While some country guidelines allow prereferral first dose administration of ACS at lower-level facilities (where BEmONC is available), implementation is limited due to challenges around identifying preterm labour, lack of knowledge about importance of prereferral dosing, and transportation issues. | [23,40,79–82] | **High confidence** | No or very minor concerns regarding coherence and adequacy yet downgraded due to minor concerns regarding relevance and moderate concerns regarding methodological limitations. |
| 7 | **Strategies to improve appropriate use** | | | |
| 7.1 | **Implementing reminder systems and educational materials**<br>Reminder systems and printed education materials (pamphlets, posters, signage) to prompt staff to prescribe and administer magnesium sulphate and ACS can facilitate appropriate use. | [23,40,63] | **High confidence** | No or very minor concerns regarding coherence and relevance yet downgraded due to minor concerns regarding methodological limitations and adequacy. |
| 7.2 | **Developing reporting indicators and audit and feedback cycles**<br>Developing and implementing key performance indicators on magnesium sulphate and ACS use for health facilities and implementing audit and feedback cycles may be enablers to encourage appropriate use. | [23,40,68,79] | **Moderate confidence** | No or very minor concerns regarding coherence but downgraded due to minor concerns regarding relevance and adequacy, as well as moderate concerns regarding methodological limitations. |
| 7.3 | **Implementing education and training for health providers**<br>Training for health providers to improve their knowledge about current research evidence, knowledge about impact of treatment on the woman and baby, and skills to administer ACS and magnesium sulphate were viewed as highly necessary and valuable. | [23,40,68,79] | **High confidence** | No or very minor concerns regarding coherence and adequacy yet downgraded due to minor concerns regarding relevance and moderate concerns regarding methodological limitations. |

(*Continued*)

**Table 3.** (Continued)

| # | Summary of qualitative review findings | Contributing qualitative studies | Overall CERQual assessment | Explanation of overall assessment |
|---|---|---|---|---|
| 1 | **Inaccurate assessment of gestational age** | | | |
| 7.4 | **Appointing "change champions"** Nominating facility-level influential obstetricians and midwives as "change champions" may help to promote and enable magnesium sulphate and ACS training and use. | [23,40,72,82] | **Moderate confidence** | No or very minor concerns regarding coherence, yet downgraded due to minor concerns regarding relevance, adequacy, and moderate concerns regarding methodological limitations. |
| 7.5 | **Multidisciplinary teamwork to improve quality of care** Multidisciplinary teamwork was highly valued by health providers to optimise ACS use, but fears, concerns, and frustrations were expressed over poor communication between the obstetric, midwifery, neonatal, and paediatric teams. Improved and standardised communication on ACS during handover and referral were highly valued but often lacking, particularly regarding whether interventions were administered yet and timing of administration. | [40,63] | **High confidence** | No or very minor concerns regarding coherence, yet downgraded due to minor concerns regarding methodological limitations, relevance, and adequacy. |
| 8 | **Women's perspectives and experiences** | | | |
| 8.1 | **Women and partners' knowledge of interventions** Women's and partners' knowledge of ACS varied across settings. In high-income countries, some women and partners understood that ACS improved fetal lung maturity but were less aware of number of doses or the name of the medication administered. In contrast, in LMIC settings, very few women or their partners were aware of ACS. | [62,80,81] | **Moderate confidence** | No or very minor concerns regarding coherence yet downgraded due to minor concerns regarding relevance and moderate concerns regarding methodological limitations as well as adequacy. |
| 8.2 | **Women learning about preterm birth management** Many women and partners first learned about preterm birth and its management (including use of tocolytics, ACS, and magnesium sulphate) during emergency situations, hindering their understanding about potential interventions and sometimes contributing to hesitancy when risks and benefits were not well understood. Some women felt that decisions concerning ACS administration should be made solely by health providers, while others felt that they needed adequate time and information to consider risks and benefits. Women felt that their knowledge and ability to make informed decisions was improved by clear communication, adequate time for discussion with their provider, as well as educational sessions and materials. | [23,40,51,62,66,67,80] | **High confidence** | No or very minor concerns regarding coherence and adequacy yet downgraded due to minor concerns regarding methodological limitations and moderate concerns regarding relevance. |
| 8.3 | **Women's experiences of and concerns about side effects** Despite personal experiences of and concerns about potential side effects of tocolytics and ACS among women in high-income countries, women mostly felt that they would take tocolytics and ACS in a future pregnancy if indicated. Some women preferred intravenous to oral tocolytics, as side effects were more consistent, with fewer "peaks and troughs" and uterine contractions. | [51,62,67] | **Moderate confidence** | No or very minor concerns regarding coherence but downgraded due to minor concerns regarding methodological limitations and adequacy, as well as moderate concerns regarding relevance. |
| 8.4 | **Women's concerns about on impact of interventions on baby** Women and partners expressed concerns about the baby's health—both from the possibility of preterm birth and from the potential impact of tocolytics on the baby. Balancing the fear of these 2 unknowns could be highly stressful, particularly as some women described feeling decreased fetal movement after tocolytic administration. | [51,66,67] | **Moderate confidence** | No or very minor concerns regarding coherence yet downgraded due to minor concerns regarding methodological limitations and adequacy, as well as moderate concerns regarding relevance. |

(*Continued*)

**Table 3.** (Continued)

| # | Summary of qualitative review findings | Contributing qualitative studies | Overall CERQual assessment | Explanation of overall assessment |
|---|---|---|---|---|
| 1 | **Inaccurate assessment of gestational age** | | | |
| 8.5 | **Regaining control and empowerment**<br>Women experiencing preterm labour placed high value on interventions that helped them to maintain autonomy and regain control over their bodies and premature labour, such as interventions that enabled them to stay out of hospital or regain mobility. These types of interventions helped to promote their freedom while giving them a sense of security regarding their baby's health. | [51] | **Low confidence** | No or very minor concerns regarding methodological limitations and coherence yet downgraded due to moderate concerns regarding relevance and serious concerns regarding adequacy. |
| 8.6 | **Trust and relationships between women and health providers**<br>Women highly valued time and space to have a 2-way conversation and build trust with their health providers to understand their condition and treatment options. While some women reported experiencing positive relationships with health providers, critical threats to building trust included insufficient health provider time due to workload, lack of continuity of carers, and perceived invalidation of women's concerns about whether they were in labour or not. | [51,62,67] | **Moderate confidence** | No or very minor concerns regarding coherence yet downgraded due to minor concerns on methodological limitations and adequacy, as well as moderate concerns regarding relevance. |
| 8.7 | **Seeking support from families and peers**<br>During preterm birth management, women leaned on their families and partners for emotional and physical support, such as motivation for staying on bedrest, general advice about pregnancy and baby health, sharing experiences, and developing coping strategies. Several women and their partners described it as challenging to ask for support from families and friends during preterm birth management, as it is less common to ask for support during pregnancy compared to after the baby is born. | [51,62,67] | **Moderate confidence** | No or very minor concerns regarding coherence yet downgraded due to minor concerns regarding methodological limitations and adequacy, as well as moderate concerns regarding relevance. |
| 8.8 | **Coping strategies—reframing experiences**<br>For women and their partners, reframing experiences of preterm birth management was critical to avoid disappointment and strengthen resolve. Reframing experiences led women and their partners to attempt to focus on positive aspects of their lives, enjoying moments with the baby, building relationships with babies, and learning to let go. | [51,67] | **Moderate confidence** | No or very minor concerns regarding methodological limitations and coherence yet downgraded due to minor concerns regarding adequacy and moderate concerns regarding relevance. |

ACS, antenatal corticosteroid; BEmONC, basic emergency obstetric and newborn care; CEmONC, comprehensive emergency obstetric and newborn care; LMIC, low- or middle-income country; PPROM, preterm prelabour rupture of membranes.

was high levels of knowledge and experience in administering the interventions, this improved implementation feasibility. Lack of knowledge or outdated knowledge were barriers to appropriate use. The key existing knowledge gaps were related to differences between research evidence and previous clinical training or experience, which sometimes involved different courses, dosing, and duration (3.1 –Moderate confidence) [23,24,40,63,80–82]. There was confusion among health providers, particularly midwives and junior doctors, about correct practices for administering ACS [40,63,82]. Some providers reported that experience and comfort in administering magnesium sulphate for preeclampsia or eclampsia can be a facilitator for using magnesium sulphate for fetal neuroprotection [82].

Quantitative evidence supported the qualitative findings around variable knowledge on ACS [53,54,59,65,73,76,78]. In India, health providers were reported to be confident in

administering ACS, despite poor score on knowledge assessment regarding the intervention [78]. Facilitators of ACS, magnesium sulphate, and tocolytics use in relation to knowledge included health providers' positive attitudes, better knowledge, exposure to trainings, conferences, guidelines, and research articles. Barriers included lack of experience in administration, misinformation about correct use, and knowledge gaps on dosing and frequency [53,54,59,65,73,76].

**Knowledge about optimal gestational age for intervention administration.** Knowledge about optimal gestational age for administration of ACS and magnesium sulphate varied across health providers, with mixed opinions about the earliest gestational age they would administer and agreement that these were challenging to have with women and families. Opinion about optimal gestational age for administration of interventions were also balanced with other factors including estimated time to birth, threatened versus imminent preterm birth, and local standards of practice (3.2 –Moderate confidence) [24,63,79,82]. Many providers perceived that ACS would be most beneficial when administered between 28 to 32 weeks, yet they were uncertain if the same benefits and no risks would be observed at earlier gestational ages [24]. A minority of obstetricians believed that there were no risks of administration and clear benefits for administration as early as 22 weeks [24]. Some neonatologists reported administering ACS up to 34 weeks, while some obstetricians reported that they would consider administering ACS up to term gestation [63]. Different opinions about optimal gestational age for ACS and magnesium sulphate may discourage providers in administering these interventions [24,63].

Quantitative evidence supported the qualitative findings about health providers knowledge about the importance of gestational age for ACS and tocolytics administration, and that knowledge about optimal gestational age range for ACS and tocolytics administration varies across settings and cadre of providers, from as early as 21 weeks to as late as 37 weeks [41,45,47,50,53,57,59,69,73–75,77].

## Perceived risks and benefits

**Uncertainties in prescribing and administering ACS for specific populations of women.** Health providers had uncertainties and lacked confidence regarding certain aspects of prescribing and administering ACS, such as whether to use repeat doses, or whether to use ACS in specific clinical situations (such as in women with diabetes, hypertension, fetal complications, maternal infection, or PPROM) (4.1 –Moderate confidence) [40,63,80]. To address these clinical uncertainties, obstetricians believed that specific guidance was needed [40,63,80]. Providers reported varied beliefs about repeat doses: While midwives expressed uncertainties and concerns regarding the evidence on benefits and risks of repeat doses, neonatologists had stronger beliefs that existing evidence supported safe administration of repeat doses [63,80].

Quantitative evidence supported the qualitative findings that health providers across settings reported variation on ACS administration practices and beliefs in certain clinical populations. Surveyed providers in quantitative studies had mixed beliefs about the benefits of administration and desired more research evidence about safety and effectiveness [42,43,46,47,52,56].

**Scepticism of the evidence base for interventions.** Health providers had mixed beliefs about the evidence supporting ACS and magnesium sulphate for fetal neuroprotection. While some providers agreed with and believed in the evidence supporting their use, others were sceptical about long-term outcomes, availability of high-quality trials, mixed evidence of effects, and sufficiency of evidence, all of which may act as barriers to use (4.2 –Moderate confidence) [23,24,40,63,82]. This scepticism was a barrier to use of ACS and magnesium sulphate,

but appeared in recent years to be dissipating. However, obstetricians, midwives, and neonatologists believed that more work was needed to increase awareness of benefits of ACS [23,40,63,82].

Quantitative evidence similarly found that while health providers agreed that ACS are beneficial, some scepticism remained due to fear of birth defects, post-administration side effects, and doubts about benefits [42–44,55,56].

**Beliefs about risks of interventions.** While many health providers believed that risks of ACS and magnesium sulphate were negligible, some had concerns about possible safety issues (particularly interactions with tocolytics, exacerbation of pulmonary oedema), low tolerance by women, long-term risks of complications for women, whether use at earlier gestational age is appropriate (<28 weeks), and risk of maternal infection (4.3 –Moderate confidence) [23,24,79,80]. These concerns were barriers to administration of magnesium sulphate and ACS [23,24].

Quantitative evidence supported the qualitative findings regarding concerns about risks after administration of ACS, magnesium sulphate, and tocolytics among health providers [42,47,56,59].

**Beliefs about risks of interventions—Interaction with tocolytics.** Some health providers believed that interaction of magnesium sulphate and ACS individually with tocolytic agents (particularly nifedipine) is associated with exacerbated adverse effects and toxicity for women. This belief may hinder administration of magnesium sulphate and ACS, in women who are also eligible for ACS and tocolytics (4.4 –Low confidence) [23,24]. No relevant quantitative evidence contributed to this finding.

**Beliefs about benefits of interventions.** Most health providers recognised the benefits of magnesium sulphate and ACS, believing that these interventions save lives, and benefits mostly outweigh risks. Women similarly believed that administration of ACS is beneficial, stressing the importance of using only when necessary and receiving information about potential side effects. In contrast, many health providers believed that tocolytics do not work and do not stop labour (4.5 –High confidence) [23,24,40,62,63,80]. Health providers expressed that an important facilitator of magnesium sulphate and ACS use is a shared belief across providers and women that these 2 interventions improve outcomes. Women's awareness of and beliefs about the benefits of magnesium sulphate and ACS are also important facilitators, as if women are familiar with the interventions, they may be more accepting of their use [23,40].

Quantitative evidence from health providers supported the qualitative findings regarding recognition of benefits of ACS and magnesium sulphate. However, quantitative evidence from women suggested that women may doubt the benefits of ACS, which can be a barrier to use [42,43,45–47,55,59,70,75].

## Barriers in administration of interventions

**Uncertainties on when to administer interventions.** The unpredictability of preterm birth, including difficulty diagnosing threatened versus imminent preterm birth, can lead to provider hesitation in administering ACS and magnesium sulphate—providers fear being held responsible or blamed for potentially unnecessary treatment. To cope with these uncertainties, providers may delay treatment, preferring a "wait and see" approach (5.1 –Moderate confidence) [23,24,63,80,82]. The "wait and see" approach can delay administration of ACS by administering tocolytics first, then waiting for 12 to 48 hours, to determine if labour decelerates before administering ACS or referring the woman [24,80]. The greater the uncertainty about the timing of preterm birth, the less likely that the providers will use ACS [23,24,63].

Quantitative evidence extended understanding of the qualitative evidence, as health providers reported using tocolytics to prolong labour to maximise the effect of ACS, and/or refer women to a higher-level facility [45,53–57,59,73].

**Time constraints and complexity in prescribing and administering.** Health providers described time constraints in prescribing and administering ACS and magnesium sulphate as a critical overarching barrier to appropriate use, due to the acute nature and time pressures of imminent preterm birth, high intensity of workload, and competing tasks. Many health providers believed that prescribing and administering magnesium sulphate is complex, as preparation takes too much time, or is difficult to "draw it all up," which could deter health providers in administering the medication when they feel under pressure (5.2 –Moderate confidence) [23,63,82]. Acknowledging the unpredictability of preterm birth and complexity of preparing magnesium sulphate regimens, health providers suggested "readymade syringes" to enable prompt administration [23].

Quantitative evidence supported the qualitative findings that insufficient time, difficulties in administering ACS, tocolytics, and magnesium sulphate, and high workloads were barriers to use [42,47,55,60,76].

**Stocking medications in maternity ward.** Maintaining consistent stock of ACS and magnesium sulphate that is readily available in the maternity ward and emergency department, and the availability of health providers who are readily able to assess women in preterm labour, was critical to ensure that women received prompt treatment (5.3 –High confidence) [23,40,68,79,80]. Where medications were stocked in the hospital pharmacy but not the maternity ward, delays in ACS and magnesium sulphate administration could occur. In some hospitals, administration of magnesium sulphate is only allowed at labour ward (not antenatal ward); therefore, women who were not in the labour ward due to overcrowding or referral issues may have delays [23].

Quantitative evidence extended the understanding of the qualitative evidence that health providers and policymakers believed that ACS and magnesium sulphate were not always available due to insufficient funding and budget allocation resulting in suboptimal procurement and distribution [22,78]. Furthermore, health providers may be comfortable prescribing dexamethasone for all women presenting with preterm labour (except for those with signs of infection), and betamethasone only to women with diabetes [22,42,44,56,60,68,77]. In some settings, dexamethasone may be the only corticosteroid available in the hospital, or the only corticosteroid stocked in the maternity setting [22,42,44,56,60,68,77].

**Regulatory policies and beliefs about prescribing and administering authority.** National-level guidance is often limited about who can prescribe and administer ACS and magnesium sulphate; where there is guidance, typically only obstetricians are authorised to prescribe and administer, while other health providers can administer under clinical oversight, but not prescribe. Many health providers (obstetricians, neonatologists, midwives) likewise believe that prescription and administration of ACS and magnesium sulphate should be prescribed and administered by obstetricians only, even though multidisciplinary decision-making was highly valued (5.4 –High Confidence) [23,40,63,79]. Health providers reported that inadequate training on safe administration of ACS at lower-level facilities is a key reason for low uptake [79] and could also be the source of unsafe use of the intervention.

Quantitative evidence extended the qualitative finding that health providers did not have clarity on who was responsible for prescribing and administering ACS and expanding prescription authority may facilitate use [22,49]. In India, decisions about administering ACS was mostly the role of doctors, but sometimes nurses or auxiliary nurse midwives [78].

## Appropriate settings for administrations

**Appropriate settings for ACS administration.** In some national guidelines and in clinical practice, administration of ACS is allowed at only at tertiary facilities where comprehensive

emergency obstetric and newborn care (CEmONC) and essential preterm newborn care interventions are available. While some country guidelines allow prereferral first dose administration of ACS at lower-level facilities (where basic emergency obstetric and newborn care (BEmONC) is available), implementation is limited due to challenges around identifying preterm labour, lack of knowledge about importance of prereferral dosing, and transportation issues (6.1 –High confidence) [23,40,79–82]. Across countries and within facilities, there is variability in the reported availability, quality, and content of preterm newborn care interventions, which complicates the determination of appropriate settings for ACS administration [79].

Quantitative evidence supported the qualitative finding that ACS and tocolytics were mostly used in higher-level health facilities and that delayed referral is a key barrier [22,59,73]. There was also variability regarding the availability of labour and newborn care facilities [78].

### Strategies to improve intervention use

**Implementing reminder systems and educational materials.**    Reminder systems and printed education materials (pamphlets, posters, signage) to prompt staff to prescribe and administer magnesium sulphate and ACS can facilitate appropriate use (7.1 –High confidence) [23,40,63]. Health providers at facilities where ACS are routinely used reported that these materials prompt them to administer to eligible women [40].

Quantitative evidence supported the qualitative finding that dissemination of educational materials about magnesium sulphate, ACS, and tocolytics are useful to health providers and can facilitate appropriate use [22,58,60,61,72,76].

**Developing reporting indicators and audit and feedback cycles.**    Developing and implementing key performance indicators on magnesium sulphate and ACS use for health facilities and implementing audit and feedback cycles may be enablers to encourage appropriate use (7.2 –Moderate confidence) [23,40,68,79]. These may be integrated as part of Health Management Information Systems and include indicators such as stock outs, proportion of women who received steroids at certain gestational ages, and proportion of women in preterm labour who receive at least 1 dose of steroids before birth [79]. Feedback on "missed opportunities" and both formal and informal discussions or "huddles" can help to identify problems and solutions and promote a quality improvement culture [23,40].

Quantitative evidence supported the qualitative finding that quality monitoring and improvement systems on ACS are varied across settings. Audit and feedback processes can help to encourage appropriate use of ACS [22,61,78].

**Implementing education and training for health providers.**    Training for health providers to improve their knowledge about current research evidence, knowledge about impact of treatment on the woman and baby, and skills to administer ACS and magnesium sulphate were viewed as highly necessary and valuable (7.3 –High confidence) [23,40,63,72,79,80,82]. Training can be delivered as both pre- and in-service training and should include information about preterm birth, and ACS roles, obstetric ultrasound training, and neonatal resuscitation [79,80].

Quantitative evidence supported qualitative evidence that education sessions, workshops, and training sessions for health providers are valuable to encourage use of magnesium sulphate and ACS [22,58,61].

**Appointing "change champions".**    Nominating facility-level influential obstetricians and midwives as "change champions" may help to promote and enable magnesium sulphate and ACS training and use (7.4 –Moderate confidence) [23,40,72,82]. "Change champions" should be comfortable listening and providing feedback to health providers about why women do and do not receive ACS [40].

Quantitative findings extended qualitative findings that involvement of community-level "change champions," such as community leaders, can facilitate ACS implementation [22,61].

**Multidisciplinary teamwork to improve quality of care.** Multidisciplinary teamwork was highly valued by health providers to optimise ACS use, but fears, concerns, and frustrations were expressed over poor communication between the obstetric, midwifery, neonatal, and paediatric teams. Improved and standardised communication on ACS during handover and referral were highly valued but often lacking, particularly regarding whether interventions were administered yet and timing of administration (7.5 –High confidence) [40,63]. Key impacts of multidisciplinary teamwork were fostering positive culture and prompting use of ACS for eligible women [40,63]. Depending on the prescribing authority in certain contexts, multidisciplinary teamwork may also encourage use by enabling more types of health providers (instead of only obstetricians) to prescribe ACS [40,63]. There was no relevant quantitative evidence about multidisciplinary teamwork.

## Women's perspectives and experiences

**Women and partners' knowledge of interventions.** Women's and partners' knowledge of ACS varied across settings. In high-income countries, some women and partners understood that ACS improved fetal lung maturity, but were less aware of number of doses or the name of the medication administered. In contrast, in LMIC settings, very few women or their partners were aware of ACS (8.1 –Moderate confidence) [62,80,81]. Women expressed feeling scared, worried, frustrated, and lacking control and autonomy when they encountered preterm labour and had limited information regarding the condition and associated interventions (ACS and tocolytics) [51,62,66]. Having limited knowledge can make women feel that they are unable to actively participate in their care through informed decision-making [51,62,66]. Therefore, when women are aware and knowledgeable about the interventions, they can more actively participate in their care, including receiving ACS [62,80,81].

Quantitative evidence supported the qualitative finding that women's knowledge about ACS and magnesium sulphate could act as a facilitator or barrier to use and that misinformation about correct use and poor understanding about benefits can be important barriers [42,56,60].

**Women learning about preterm birth management.** Many women and partners first learned about preterm birth and its management (including use of tocolytics, ACS, and magnesium sulphate) during emergency situations, hindering their understanding about potential interventions and sometimes contributing to hesitancy when risks and benefits were not well understood. Some women felt that decisions concerning ACS administration should be made solely by health providers, while others felt that they needed adequate time and information to consider risks and benefits. Women felt that their knowledge and ability to make informed decisions was improved by clear communication, adequate time for discussion with their provider, as well as educational sessions and materials (8.2 –High confidence) [23,40,51,62,66,67,80]. Some women preferred to learn more about preterm birth and preterm birth management earlier in pregnancy—for example, at antenatal care—to allow more time to understand what may happen and how it may be managed [51,62]. Women with previous experience of preterm labour reported increased awareness about the likelihood of recurrence of preterm labour and knowledge of management options, which may provide them with greater confidence in making informed choices and negotiating their care. Similarly, family members of women with previous preterm birth also reported experiencing less worry regarding tocolytics' impact on the baby compared with the woman's previous pregnancy [51].

Quantitative evidence supported the qualitative findings that women typically learn about ACS from their health providers and that some women may not accept ACS and magnesium sulphate due to fears about injections or disapproval from their husband or partner [42,52].

**Women's experiences of and concerns about side effects.** Despite personal experiences of and concerns about potential side effects of tocolytics and ACS among women in high-income countries, women mostly felt that they would take tocolytics and ACS in a future pregnancy if indicated. Some women preferred intravenous to oral tocolytics, as side effects were more consistent, with fewer "peaks and troughs" and uterine contractions (8.3 –Moderate confidence) [51,62,67]. Many women experienced side effects from oral or intravenous tocolytics (terbutaline, magnesium sulphate, ritodrine), including nausea, vomiting, weakness, dizziness, fatigue, double vision, lack of appetite, and tachycardia. Some also experienced sleep deprivation due to the need to take oral medications every few hours. Intravenous administration limited women's mobility and made basic tasks more complicated and reduced their autonomy [51]. No quantitative evidence supported this theme.

**Women's concerns about on impact of interventions on baby.** Women and partners expressed concerns about the baby's health—both from the possibility of preterm birth and from the potential impact of tocolytics on the baby. Balancing the fear of these 2 unknowns could be highly stressful, particularly as some women described feeling decreased fetal movement after tocolytic administration (8.4 –Moderate confidence) [51,66,67]. Some women perceived decreased fetal movement when administered with intravenous magnesium sulphate, which prompted them to stop treatment [51,66]. No quantitative evidence supported this theme.

**Regaining control and empowerment.** Women experiencing preterm labour placed high value on interventions that helped them to maintain autonomy and regain control over their bodies and premature labour, such as interventions that enabled them to stay out of hospital or regain mobility. These types of interventions helped to promote their freedom while giving them a sense of security regarding their baby's health (8.5 –Low confidence) [51]. One intervention that women mention was terbutaline pump therapy that women can administer independently at home [51].

**Trust and relationships between women and health providers.** Women highly valued time and space to have a 2-way conversation and build trust with their health providers to understand their condition and treatment options. While some women reported experiencing positive relationships with health providers, critical threats to building trust included insufficient health provider time due to workload, lack of continuity of carers, and perceived invalidation of women's concerns about whether they were in labour or not (8.6 –Moderate confidence) [51,62,67]. Both women and partners described how relationships with health providers could break down, resulting in women feeling neglected and not understanding why certain procedures were conducted and feeling that there was limited recourse to discuss their experience with their health providers [51,67].

**Seeking support from families and peers.** During preterm birth management, women leaned on their families and partners for emotional and physical support, such as motivation for staying on bedrest, general advice about pregnancy and baby health, sharing experiences, and developing coping strategies. Several women and their partners described it as challenging to ask for support from families and friends during preterm birth management, as it is less common to ask for support during pregnancy compared to after the baby is born (8.7 –Moderate confidence) [51,62,67]. While obtaining support was considered important, people in women's social networks sometimes made negative comments about whether interventions were safe for the baby, which could invoke guilt [51,62]. Some women found peer support from other women undergoing preterm birth management (tocolytics) was helpful for emotional support from someone undergoing a similar procedure at the same time [51].

**Coping strategies—Reframing experiences.** For women and their partners, reframing experiences of preterm birth management was critical to avoid disappointment and strengthen resolve. Reframing experiences led women and their partners to attempt to focus on positive aspects of their lives, enjoying moments with the baby, building relationships with babies, and learning to let go (8.8 –Moderate confidence) [51,67]. Women reported reframing experience through setting goals and celebrating, looking to religion, and creating routine, while partners reported reframing experience by minimising their expectations to avoid disappointment [51,67].

## Preterm premature rupture of membranes (PPROM) management with antibiotics

There were no qualitative studies contributing evidence on use of antibiotics for PPROM; however, quantitative studies found that prescribing antibiotics for women with PPROM was common [48,57,73]. While some providers reported using antibiotics for PPROM due to evidence of benefit, national guidance, and as Group B Streptococcal Disease (GBS) prophylaxis, some providers reported non-use due to the perception of inconclusive evidence [71]. Antibiotic regimens were highly variable across settings (see S6 Appendix) [48,57,70,71,73].

## Mapping to behaviour change frameworks

We mapped facilitators and barriers from the qualitative and quantitative synthesis to the TDF [25] and COM-B frameworks [26] to understand how addressing factors affecting implementation may influence appropriate use of the interventions by providers and acceptability of interventions use by women. This approach can also help to identify implementation strategies for future research on scaling up appropriate use of the interventions. Figs 3 and 4 present the mapping of factors affecting health providers' appropriate use of interventions for preterm birth management. From the barriers and facilitators identified on each of the 3 COM-B domains, capability, motivation, and opportunity, we can see that in order to improve health providers' capability, implementation of training, education materials, reminder system, as well as audit and feedback are needed. Motivation of health providers can be leveraged through "change champions" and improved and standardised communication between health providers. Lastly, opportunity can be improved by disseminating consistent, detailed, and clear clinical practice guidelines and by ensuring adequate human and nonhuman resources (ultrasound dating, medication stock, availability of labour and preterm birth interventions, referral system) needed for appropriate use.

Fig 3 mapped facilitators and barriers affecting appropriate use of ACS, tocolytics, magnesium sulphate, and antibiotics by health providers. Red represents barriers of use, purple as facilitators of use, yellow as mixed evidence, and grey as no evidence available. Across the interventions, factors affecting use are homogeneous: When it is a barrier in one intervention, it is also a barrier in other intervention. The exception, however, can be seen on health providers practice in implementing "wait and see" approach before ACS administration, which serve as a barrier for ACS use, yet a facilitator for tocolytics. From this figure, we can also see that less is known about tocolytics and antibiotics from providers' side.

Fig 4 mapped the factors listed on Fig 3 to a COM-B model where we can clearly see the facilitators and barriers of appropriate use of ACS, tocolytics, magnesium sulphate, and antibiotics across 3 main domains that needs to be present for the behaviour change to occur: capability, motivation, and opportunity. As barriers are identified across the 3 domains, it is important that the barriers from each of the domains are addressed by implementing identified strategies when aiming to improve health providers appropriate use of these interventions. The barriers and facilitators to improve appropriate use can be clearly seen. To improve

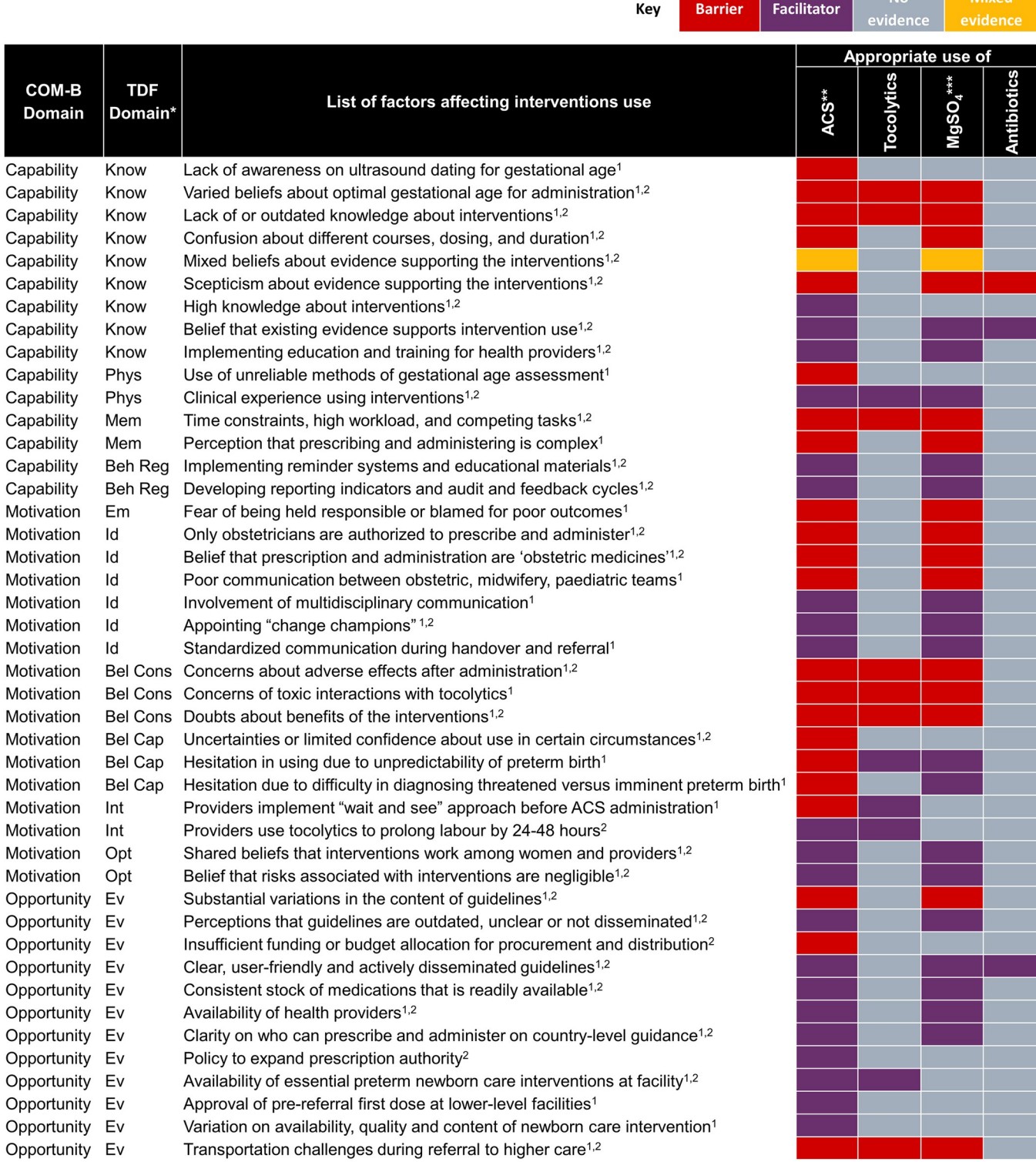

**Fig 3. TDF and COM-B mapping of factors affecting appropriate use of ACS, tocolytics, magnesium sulphate, and antibiotics by health providers.**
*Know, Knowledge; Phys, Physical skills; Mem, Memory, attention, and decision processes; Beh Reg, Behavioural regulation; Em, Emotion; Id, Social/professional role and identity; Bel Cons, Belief about consequences; Bel Cap, Belief about capabilities; Int, Intentions; Opt, Optimisms; Ev, Environmental context and resources. **ACS; ***Magnesium sulphate. [1]Factor identified from qualitative evidence. [2]Factor identified from quantitative evidence.

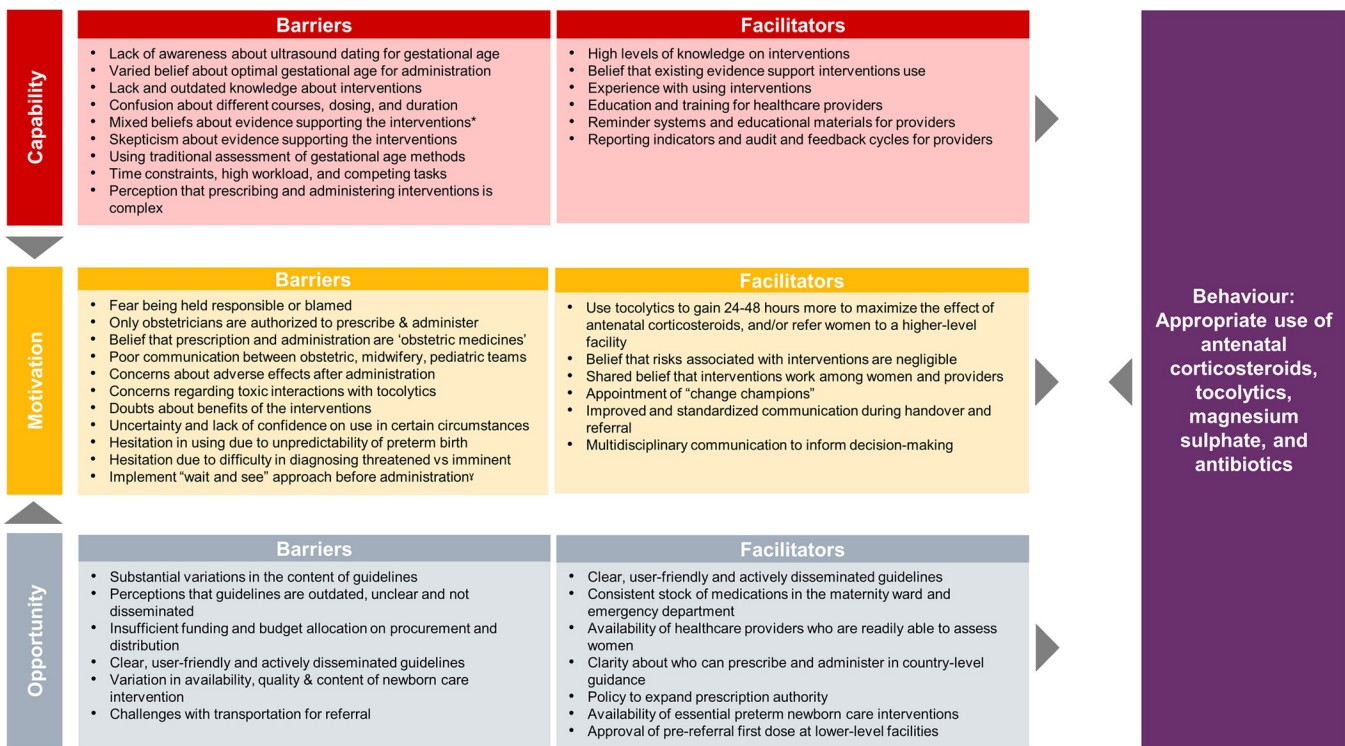

**Fig 4. Mapping factors affecting health providers' appropriate use of interventions for preterm birth management using COM-B.** * = mixed evidence, could be facilitators and barriers; γ = barriers for ACSs and magnesium sulphate, yet facilitators for tocolytics.

capability, implementation of education, training, reminder system, education materials as well as audit and feedback may be needed. Motivation of health providers can be leveraged through appointment "change champion," improved and standardised communication, as well as multidisciplinary communication. Opportunity can also be improved through dissemination of clear guidelines, ensure medication stock as well as adequacy of human and nonhuman resources (i.e., availability of preterm birth interventions) needed for appropriate use.

Fig 5 presents the mapping of factors affecting women's acceptability to receive interventions for preterm birth management. Women may be more likely to accept interventions for preterm birth management when they have access to education sessions and materials to support decision-making (capability), when benefits are clear and reinforced (motivation), and they are appropriately supported by health providers and their social networks (opportunity). Similar to factors affecting appropriate use by health providers, factors affecting acceptability of women are also homogeneous across interventions: When it is a barrier in one intervention, it is also a barrier in other intervention.

Fig 6 mapped the factors listed on Fig 5 to a COM-B model where we can clearly see the facilitators and barriers of acceptability to use ACS, tocolytics, magnesium sulphate, and antibiotics by women across 3 main domains that needs to be present for the behaviour change to occur: capability, motivation, and opportunity. To improve capability, implementation of education sessions and materials for women and families are needed. Motivation of women can be leveraged by emphasising the benefits of the interventions and ensure that women actively participate and in control for their treatment. In terms of opportunity, ensuring women having adequate support from health providers as well as family members are important in improving acceptability to the interventions.

Mapping to the behaviour change frameworks facilitated understanding of how the interplay between facilitators and barriers across these domains influenced the intended behaviour (appropriate use of ACS, tocolytics, magnesium sulphate, and antibiotics) and therefore is a starting place for developing implementation strategies to reinforce facilitators and address barriers. We hypothesise that when facilitators are reinforced and barriers are removed, this will ultimately lead to health providers' appropriate use of interventions for preterm birth management and women's acceptability of these interventions.

## Discussion

Our review demonstrates the complexity of factors influencing the use of ACS, tocolytics, magnesium sulphate, and antibiotics for PPROM globally. We found 46 studies, mostly from high- and middle-income countries and mostly from health providers' perspectives. Limited availability of ultrasound gestational age dating, mixed knowledge about the effectiveness and safety of the interventions, and wrong beliefs about optimal gestational age for administration are critical barriers. Across contexts, wide variability in guidelines exists in terms of what constitutes imminence of preterm birth, gestational age criteria, maternal infections that contraindicate use, competency and authority regulated for prescription and administration, and enabling environments for administration. The inherently unpredictable nature of spontaneous preterm birth and complexity in administering these interventions complicates decision-making and implementation. Health system challenges further complicate appropriate use, such as maintenance of adequate stock, appropriate human resources for ultrasound dating, prescription and administration of interventions, and inconsistencies in availability, quality, and content of preterm labour and newborn care environments. Women also reported hesitancy in utilising interventions as they mostly learned about it during an emergency. Despite these challenges, appropriate education for health providers and women, reminder systems, audit and feedback, change champions, and multidisciplinary teamwork may be critical levers to promote appropriate use.

Accurate gestational age assessment using ultrasound dating is critical in supporting time-sensitive interventions for preterm birth management. WHO recommends early ultrasound dating before 24 weeks gestational age to detect potential pregnancy complications and improve women's pregnancy experiences [83]. However, our review shows that ultrasound dating is relatively scarce in LMICs [78], and inaccurate methods are still used, such as last menstrual period, fundal height, and timing of first antenatal visit [68,80,81]. Many community workers are unaware on the role of ultrasound dating in pregnancy [68,80,81], and ultrasound machines may only be available at higher level hospitals, which may hinder appropriate use of the interventions [78]. Programme implementers should ensure that low-resource settings have the resources and skills to provide ultrasound dating before implementing preterm birth interventions to ensure safety and minimise harm. Innovations in ultrasound technology such as handheld or portable ultrasound devices have been developed and may be particularly useful to improve and scale up basic ultrasound services in LMIC settings.

Provider knowledge about the interventions was a facilitator to use; however, we observed variable knowledge and beliefs about optimal gestational age and specific populations of women in which interventions can be administered, which served as barriers to use. Variable knowledge and beliefs may reflect inconsistencies in the content of guidelines disseminated regarding these interventions. For example, administration of magnesium sulphate is recommended to be administered to eligible women before 32 weeks gestational age by WHO [12], but this gestational age ranges from 24 to 29$^{+6}$ weeks in guidelines issued by UK National Institute for Health and Care Excellence [18]. Furthermore, some guidelines lack critical

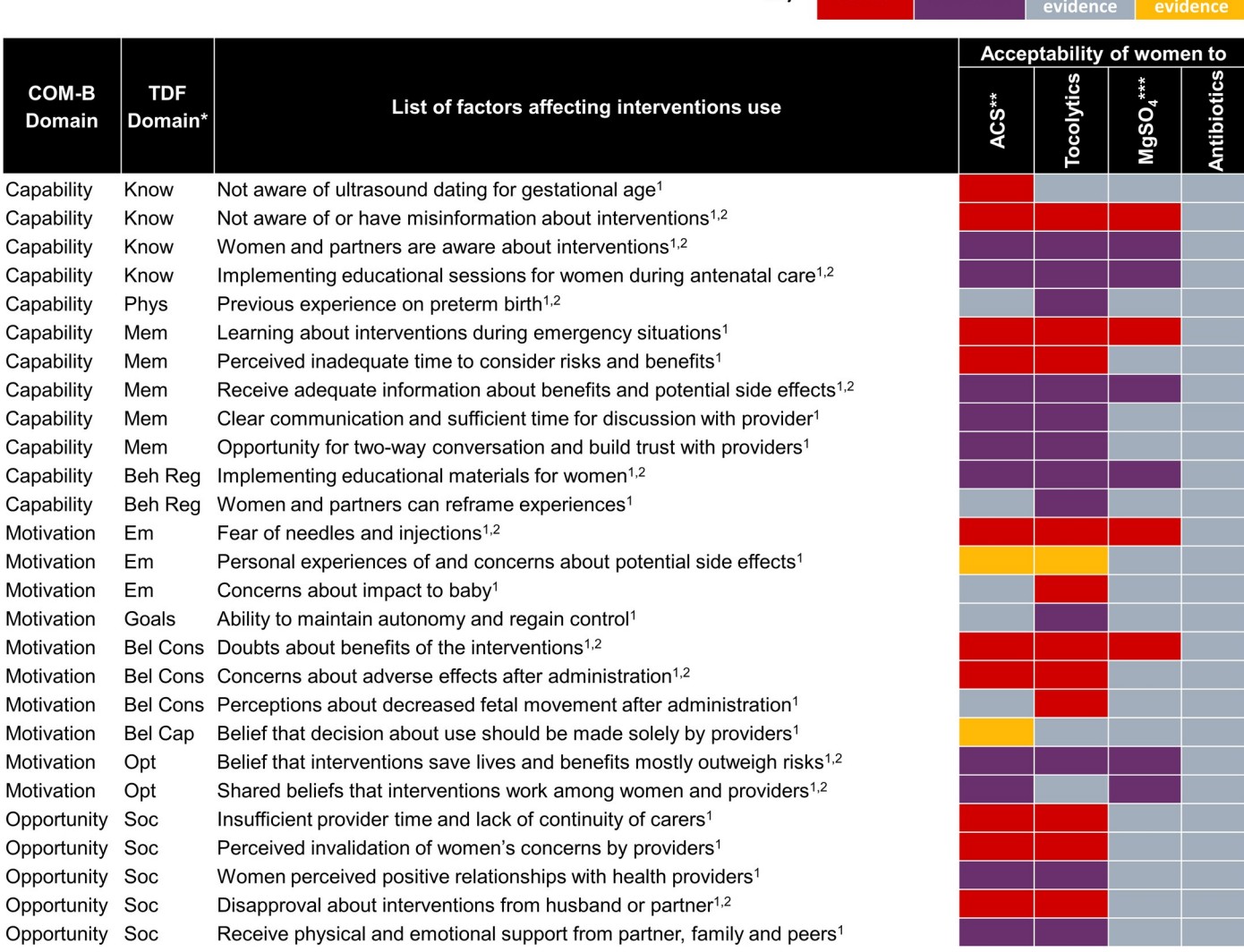

**Fig 5. TDF and COM-B mapping on factors affecting acceptability of women to use ACS, tocolytics, magnesium sulphate, and antibiotics.** *Know, Knowledge; Phys, Physical skills; Mem, Memory, attention, and decision processes; Beh Reg, Behavioural regulation; Em, Emotion; Id, Social/professional role and identity; Bel Cons, Belief about consequences; Bel Cap, Belief about capabilities; Int, Intentions; Opt, Optimisms; Ev, Environmental context and resources. **ACS; ***Magnesium sulphate. [1]Factor identified from qualitative evidence. [2]Factor identified from quantitative evidence.

information, such as range of recommended gestational age, prescribing authority or contraindications of ACS use when infection is present [79]. Guideline variation is in part due to the limited evidence base for several important questions regarding populations and optimal timing of administration. More work is needed to ensure detailed, clear, and consistent information about interventions is present in national guidelines and facility-level clinical protocols and to ensure that this guidance is actively disseminated.

Women's acceptability to the interventions are also critical to address barriers of implementation. Many clinical interventions often unintentionally leave women to be part of the narrative in ensuring use, yet results of this review shows that women often feel hesitate in using the interventions as they are unfamiliar about the interventions and that they mostly learn about the it during emergency situations [23,40,51,62,66,67,80]. In practice, women may not be educated about preterm birth unless and until they are at high risk, hence why women who have

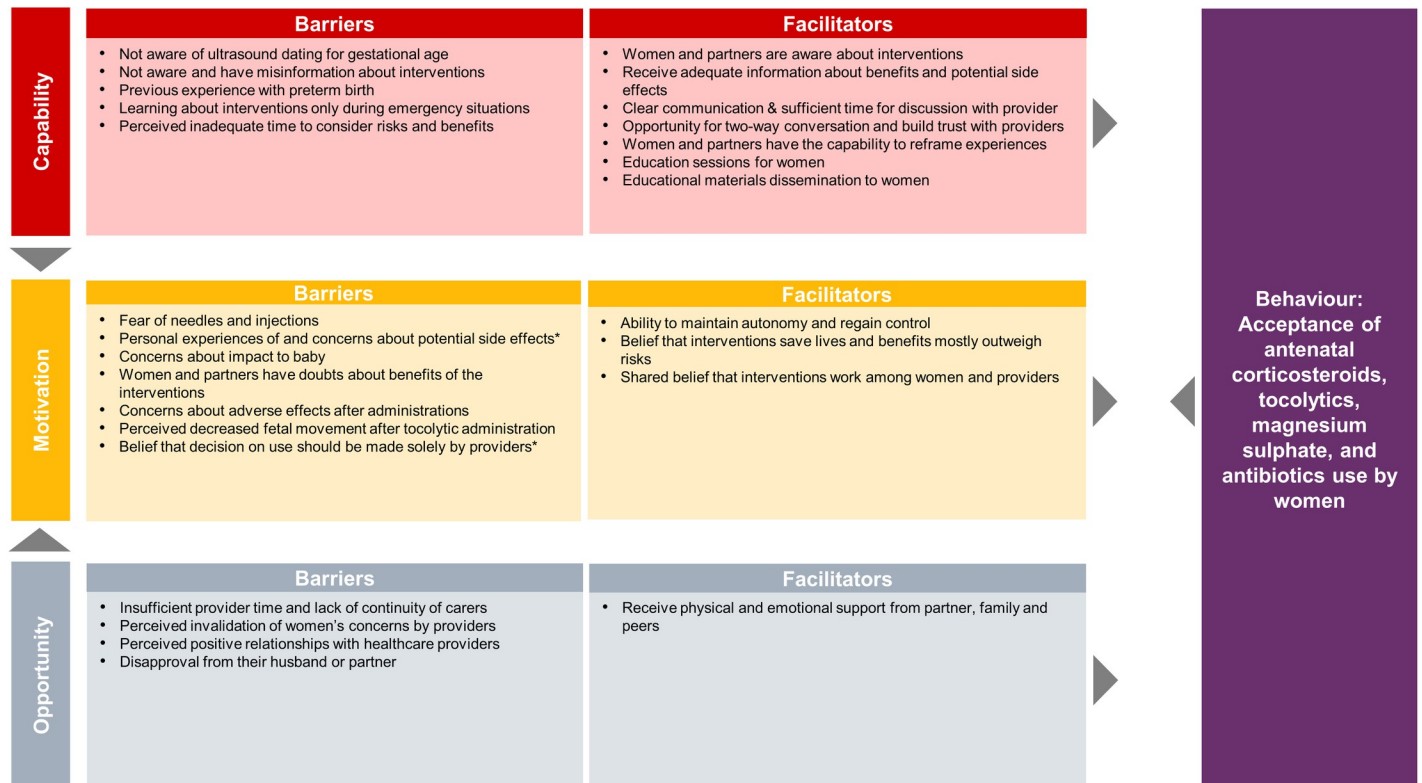

**Fig 6. Mapping factors affecting women's acceptability to receive interventions for preterm birth management using COM-B.** * = mixed evidence, could be facilitators and barriers.

experienced preterm birth in a previous pregnancy report better knowledge and feeling more confident in decision-making [51].

The TDF and COM-B mapping in our review can be used by researchers and programme implementers to inform the development of implementation models for optimal use of preterm birth management interventions in LMIC settings. Assessing the extent to which the barriers and facilitators identified in our review are potential implementation challenges in different settings is a useful starting point for formative research to scale up these preterm birth management interventions. Table 4 presents a list of questions derived from our findings and may help programme managers, policymakers, researchers, and other key stakeholders to identify and address factors that may affect implementation and scale-up of ACS, tocolytics, magnesium sulphate for fetal neuroprotection, and antibiotics for PPROM.

Most included studies were from high-income countries, which may affect the transferability of these findings to LMIC settings. We did not observe substantial differences between studies coming from different country income levels, thus we did not expect there would be much difference in the views of health providers' and women in LMICs. However, this limitation highlights the importance of primary formative research and evaluation in LMICs about implementation and scale-up of preterm birth management. More work is urgently needed to implement these 4 interventions for preterm birth management in LMIC settings, where 80% of global preterm births occur, and to evaluate implementation strategies to share learnings across contexts [2]. The scope of our review meant that we did not include studies that aimed to promote early antenatal care or birth in health facilities, or optimising care for the woman and newborn in the postpartum period. Understanding interventions during these periods is

**Table 4. Implications for practice.** This table presents a list of questions derived from our findings and may help programme managers, policymakers, researchers, and other key stakeholders to identify and address factors that may affect implementation and scale-up of ACS, tocolytics, magnesium sulphate for fetal neuroprotection, and antibiotics for PPROM. Assessing the extent to which the barriers and facilitators identified in our review are potential implementation challenges in different settings is a useful starting point for formative research to scale up these preterm birth management interventions.

| Domain | List of questions |
|---|---|
| **Accurate assessment of gestational age** | 1. Are health providers aware of ultrasound dating in the management of preterm birth?<br>2. Is an ultrasound equipment available at the health facility, and is there consistent coverage of skilled sonographers or health providers in ultrasound dating?<br>3. Is early trimester ultrasound as recommended by WHO routinely practiced? |
| **Guidelines and perceived knowledge** | 4. Are providers aware of the benefits of the ACS, tocolytics, magnesium sulphate for fetal neuroprotection, and antibiotics for PPROM for preterm birth management?<br>5. Do providers have any scepticism or concerns about adverse effects of preterm birth management that can be addressed?<br>6. Do national guidelines have clear criteria on appropriate use of the ACS, tocolytics, magnesium sulphate for fetal neuroprotection, and antibiotics for PPROM, including the following:<br> a. Guidance on assessing imminent preterm birth?<br> b. Appropriate gestational criteria for administration and determination of appropriate gestational age?<br> c. Determination of signs of maternal infection and contraindication of use when maternal infection is present?<br> d. Minimum standards for appropriate facilities to administer interventions, including essential newborn care?<br> e. Which cadre of providers can prescribe and administer the interventions?<br> f. Specific populations in which the interventions can or cannot be administered?<br>7. Are guidelines and clinical protocols on of ACS, tocolytics, magnesium sulphate for fetal neuroprotection, and antibiotics for PPROM consistent between WHO, national, and facility levels? |
| **Administration of interventions** | 8. Can administration of ACS, tocolytics, magnesium sulphate for fetal neuroprotection, and antibiotics for PPROM be simplified through packaged or ready-to-use doses?<br>9. Are relevant drugs readily available in the antenatal, labour, and emergency wards?<br>10. Is there sufficient funding and budget allocation to ensure continuous procurement and distribution of ACS, tocolytics, magnesium sulphate for fetal neuroprotection, and antibiotics for PPROM?<br>11. Has communication about administration and dosing during handover and referral been standardised? |
| **Appropriate settings for administration** | 12. Do facilities administering ACS, tocolytics, magnesium sulphate for fetal neuroprotection, and antibiotics for PPROM have adequate childbirth and preterm newborn care environments (such as resuscitation, thermal care, feeding support, infection treatment, and safe oxygen use)?<br>13. Can diagnosis of imminent preterm birth can be made lower-level health facility?<br>14. Can a prereferral dose be administered at a lower-level health facility?<br>15. Can improvements be made to the referral system, including transport? |
| **Strategies to improve use** | 16. Have health providers received sufficient training on use of ACS, tocolytics, magnesium sulphate for fetal neuroprotection, and antibiotics for PPROM?<br>17. Are there available reminder systems and educational materials on ACS, tocolytics, magnesium sulphate for fetal neuroprotection, and antibiotics for PPROM available and accessible?<br>18. Are key performance indicators and audit and feedback available for ACS, tocolytics, magnesium sulphate for fetal neuroprotection, and antibiotics for PPROM?<br>19. Have change champions or opinion leaders to promote use of ACS, tocolytics, magnesium sulphate for fetal neuroprotection, and antibiotics for PPROM been appointed at health facility? |

*(Continued)*

**Table 4.** (Continued)

| Domain | List of questions |
|---|---|
| **Women's acceptability on using interventions** | 20. Do women and partners receive education and educational materials on signs of preterm birth and preterm birth management early in pregnancy? 21. Do women have sufficient time and opportunity to discuss preterm birth management plans with health providers? |

ACS, antenatal corticosteroid; PPROM, preterm prelabour rupture of membranes; WHO, World Health Organisation.

critical to improve early identification of threatened preterm birth and improve care of small or sick newborns. Lastly, ACS effectiveness and safety in LMIC settings has only just been confirmed with the WHO ACTION-1 trial published in 2020 [84,85]; therefore, the impact of more recent evidence may not have been reflected in the studies included in this review.

Despite these limitations, to the best of our knowledge, this is the first systematic review aiming to understand factors affecting implementation of key preterm birth management interventions globally: ACS, tocolytics, magnesium sulphate for neuroprotection, and antibiotics for PPROM. Including 4 preterm birth management interventions allowed for opportunity to explore the interconnection of preterm birth management plans, rather than focusing on single interventions. The mixed-methods approach also ensures that we have an in-depth understanding of the factors of intervention use across different type of evidence. Using TDF and COM-B behaviour change frameworks enabled us to identify critical levers and implementation challenges that could be addressed to optimise future implementation of these interventions, including in LMIC settings. Policymakers, researchers, and implementers should consider these facilitators, barriers, and potential strategies when formulating policies and planning the implementation or scale-up of these interventions.

## Supporting information

**S1 Appendix. PRISMA reporting checklist.**
(PDF)

**S2 Appendix. ENTREQ reporting checklist.**
(PDF)

**S3 Appendix. Search strategies.**
(PDF)

**S4 Appendix. Critical appraisal.**
(PDF)

**S5 Appendix. GRADE-CERQual evidence profile.**
(PDF)

**S6 Appendix. Summary of quantitative findings.**
(PDF)

**S7 Appendix. Summary of study designs and type of interventions.**
(PDF)

**S8 Appendix. Development of themes.**
(PDF)

## Acknowledgments

We extend our thanks to Anayda Portela (Department of Maternal, Newborn, Child and Adolescent Health, World Health Organisation) for her valuable input into the review protocol and initial analysis, Jim Berryman (Brownless Medical Library, Faculty of Medicine, Dentistry and Health Sciences, The University of Melbourne) for defining and implementing the search strategy, and Weilong Cheng (Centre for Epidemiology and Biostatistics, Melbourne School of Population and Global Health, The University of Melbourne) in screening and translating of studies published in Mandarin.

The contents of this publication are the responsibility of the authors and do not reflect the views of the UNDP/UNFPA/UNICEF/WHO/World Bank Special Programme of Research, Development and Research Training in Human Reproduction (HRP), World Health Organisation.

## Author Contributions

**Conceptualization:** Joshua P. Vogel, Olufemi T. Oladapo, Meghan A. Bohren.

**Data curation:** Rana Islamiah Zahroh, Meghan A. Bohren.

**Formal analysis:** Rana Islamiah Zahroh, Alya Hazfiarini, Katherine E. Eddy, Joshua P. Vogel, Özge Tunçalp, Nicole Minckas, Fernando Althabe, Olufemi T. Oladapo, Meghan A. Bohren.

**Funding acquisition:** Joshua P. Vogel, Olufemi T. Oladapo, Meghan A. Bohren.

**Investigation:** Rana Islamiah Zahroh, Alya Hazfiarini, Katherine E. Eddy, Joshua P. Vogel, Özge Tunçalp, Nicole Minckas, Fernando Althabe, Olufemi T. Oladapo, Meghan A. Bohren.

**Methodology:** Rana Islamiah Zahroh, Alya Hazfiarini, Katherine E. Eddy, Joshua P. Vogel, Özge Tunçalp, Nicole Minckas, Fernando Althabe, Olufemi T. Oladapo, Meghan A. Bohren.

**Project administration:** Rana Islamiah Zahroh, Meghan A. Bohren.

**Resources:** Meghan A. Bohren.

**Software:** Rana Islamiah Zahroh, Meghan A. Bohren.

**Supervision:** Meghan A. Bohren.

**Validation:** Meghan A. Bohren.

**Visualization:** Rana Islamiah Zahroh, Meghan A. Bohren.

**Writing – original draft:** Rana Islamiah Zahroh, Alya Hazfiarini, Meghan A. Bohren.

**Writing – review & editing:** Rana Islamiah Zahroh, Alya Hazfiarini, Katherine E. Eddy, Joshua P. Vogel, Özge Tunçalp, Nicole Minckas, Fernando Althabe, Olufemi T. Oladapo, Meghan A. Bohren.

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
