## [Editor Report · Decision Letter 0]

16 Dec 2021

Dear Dr Bohren, 

Thank you for submitting your manuscript entitled "Factors influencing appropriate use of interventions for management of women experiencing preterm birth: a mixed-methods systematic review and narrative synthesis" for consideration by PLOS Medicine.

Your manuscript has now been evaluated by the PLOS Medicine editorial staff and I am writing to let you know that we would like to send your submission out for external peer review.

Please re-submit your manuscript within two working days.

Kind regards,

Louise Gaynor-Brook, MBBS PhD

PLOS Medicine

---

## [Decision Letter · Decision Letter 1]

13 May 2022

Dear Dr. Bohren,

Thank you very much for submitting your manuscript "Factors influencing appropriate use of interventions for management of women experiencing preterm birth: a mixed-methods systematic review and narrative synthesis" (PMEDICINE-D-21-05120R1) for consideration at PLOS Medicine. 

Your paper was evaluated by three independent reviewers, including a statistical reviewer, and discussed among all the editors here and with an academic editor with relevant expertise. The reviews are appended at the bottom of this email and any accompanying reviewer attachments can be seen via the link below:

[LINK]

In light of these reviews, I am afraid that we will not be able to accept the manuscript for publication in the journal in its current form, but we would like to consider a revised version that addresses the reviewers' and editors' comments. Obviously we cannot make any decision about publication until we have seen the revised manuscript and your response, and we plan to seek re-review by one or more of the reviewers. 

We expect to receive your revised manuscript by May 27 2022 11:59PM. Please email us (plosmedicine@plos.org) if you have any questions or concerns.

We look forward to receiving your revised manuscript. 

Sincerely,

Louise Gaynor-Brook, MBBS PhD

PLOS Medicine

plosmedicine.org

Comments from the Academic Editors:

I disagree with reviewer 2's perspective re Ultrasound assessment of gestational age. I do not see this recommendation as problematic in this context and thinks the authors discuss this appropriately.

What geographies/studies might be associated with the different barrier types? For example, it would be better to say that the limited diagnostics for gestational age means that rural areas have inability to determine gestational age, rather than to say that gestational age assessment is a barrier. 

Maybe there could be a comment on what might be missed and why/how. This is a window not of exactly where the problems are but where the literature has located the problems. Often, for example, structural things are harder to study and get swept under the carpet.

General comments:

Throughout the paper, please adapt reference call-outs to the following style: "... 24 to 35 weeks gestational age [12,18,19]." (noting the absence of spaces within the square brackets).

Please remove formatting such as emboldening and italicisation in the main text of your manuscript (particularly in the Results sections) 

Abstract:

Please report your abstract according to PRISMA for abstracts, following the PLOS Medicine abstract structure (Background, Methods and Findings, Conclusions) - http://www.plosmedicine.org/article/info:doi/10.1371/journal.pmed.1001419

Abstract Methods and Findings:

Please provide the number of studies included/number of participants, types of study designs included (e.g. RCTs, cohort studies, etc), eligibility criteria, and synthesis/appraisal methods (including evaluation of study quality and risk of bias). 

In the last sentence of the Abstract Methods and Findings section, please describe 2-3 of the main limitations of the study's methodology.

Abstract Conclusions:

Please begin your Abstract Conclusions with "In this study, we observed ..." or similar, to summarize the main findings from your study, without overstating your conclusions. Please emphasize what is new and address the implications of your study, being careful to avoid assertions of primacy. 

Author Summary:

In the final bullet point of ‘What Do These Findings Mean?’, please describe the main limitations of the study in non-technical language.

Introduction:

If there has been a systematic review of the evidence related to your study (or you have conducted one), please refer to and reference that review. 

Methods:

Please state early in the Methods section whether any reported analyses differed from those that were planned in your PROSPERO protocol. Changes in the analysis, including those made in response to peer review comments, should be identified as such in the Methods section of the paper, with rationale. If a reported analysis was performed based on an interesting but unanticipated pattern in the data, please be clear that the analysis was data-driven.

Thank you for providing a completed PRISMA checklist. Please add the following statement to the Methods: "This study is reported as per the Preferred Reporting Items for Systematic Reviews and Meta-Analyses (PRISMA) guideline (S1 Checklist).” 

Please consider including PubMed in your search.

Please update your search to the present time. We require that SRs are updated to within roughly 6 months of the expected publication date.

Results: 

Please include a table showing the characteristics of the included studies, including data from the original studies on the participants, interventions and outcomes, which should be incorporated into the main paper (this is currently Appendix S7). 

Discussion:

Please present and organize the Discussion as follows: a short, clear summary of the article's findings; what the study adds to existing research and where and why the results may differ from previous research; strengths and limitations of the study; implications and next steps for research, clinical practice, and/or public policy; one-paragraph conclusion.

Please remove all subheadings within your Discussion e.g. Summary of main results

Please move the section on ‘Implications for practice’ in its current form to the supplementary information, and discuss the implications of your findings in prose form within the main text of the Discussion 

Line 760 - please temper assertions of primacy by adding ‘to the best of our knowledge’ or similar 

Figures:

Please define all abbreviations used in the figure legend of each figure.

Please consider avoiding the use of red and green in order to make your figure more accessible to those with colour blindness.

Tables:

Please define all abbreviations used in the table legend of each table.

References:

Please ensure that journal name abbreviations match those found in the National Center for Biotechnology Information (NCBI) databases (http://www.ncbi.nlm.nih.gov/nlmcatalog/journals), and are appropriately and consistently formatted and capitalised e.g. refs 3, 5, 7, etc.

Please also see https://journals.plos.org/plosmedicine/s/submission-guidelines#loc-references for further details on reference formatting. 

Where website addresses are cited, please specify the date of access. 

Supplementary files: 

Please see https://journals.plos.org/plosmedicine/s/supporting-information for our supporting information guidelines. 

Comments from the reviewers:

Reviewer #1: This is a systematic review of both quantitative and qualitative studies to identify factors influencing appropriate use of interventions for preterm birth. The study, ambitiously has five objectives, to 1) explore perceptions, preferences and experiences of women, partners, health providers and other relevant stakeholders on the use of four preterm birth interventions; 2) explore how health workers identify women at risk of preterm birth, including assessment of gestational age, identifying signs of maternal infection, and recognizing risk of preterm birth; 3) identify factors affecting administration and duration of exposure of the four interventions; 4) explore whether the factors affecting appropriate use differ across types of health facilities; and 5) use Theoretical Domains Framework (TDF) and Capability, Opportunity, and Motivation (COM-B) model of behavior change to explore potential strategies in improving appropriate use and scale up of the four interventions. Overall, I think this is an ambitious and comprehensive study trying to provide useful evidence on barriers and facilitators of interventions of preterm birth and give suggestion on implementation strategies. Here are my specific comments:

1. Line 116-117: Why did you only include studies with cross-sectional and mixed-methods approaches for data collection? Why not consider studies collecting data using other methods, such as surveillance, registration, or hospital/facility record? Also, Why not included studies of interventions that increase the use of those preventative measures. Those interventions may target specific factors that prohibit of promote use of the four interventions.

2. Line 127-129: Why did you only included studies exploring perspectives of people? Why not studies exploring factors measured more objectively such as level of facility, quantity, and quality of health personnel and other resources?

3. Line 133: Why did you not use PubMed?

4. Line 137: how did you decide to only look at the first 10 pages?

5. Line 173: How did you select the initial 6 studies for developing the codebook? What are the specific criteria?

6. Line 176-178: The methods of synthesizing qualitative data need to be reviewed by an expert of qualitative analysis.

7. Line 179-181: Why not the other why around? Why qualitative data were not used to extend quantitative evidence? To me, qualitative data should be much broader than quantitative data, especially given that you have more quantitative studies (32) than qualitative studies (11).

8. Line 179-187: I think you may need some more explanation on how you mapped the quantitative data and qualitative findings and on how you mapped the findings to the TDF and COM-B.

9. Line 188: provides.

10. Line 208-218: may put some of the details to the appendix here.

11. Lastly, I recommend the study be reviewed by experts on qualitative analysis and on preterm births.

Reviewer #2: The authors have done an interesting review, that is much needed. The authors provide impressive and interesting tables and figures. 

The manuscript is dense, and as I understand it is based on work done for a WHO recommendation. However, if this is to be an article, rather than a report, I find the text in the results section difficult to go through. There is quite some repetition and I wonder if this could be better covered by the tables provided, with the results section in the text to be reserved for contextualising these results. 

I have one major concern, which might be resolved by simple changes in language - but might also more complex as this has to do with the near absence of focus on context, which is only given some attention in the discussion. The authors have a clear focus on the importance of GA dating via ultrasound, and in line 643 other methods of determining GA are deemed Inaccurate. Yet, the reality is that in the majority of countries in LMIC setting this is the Only method available for GA dating. And while this might indeed be LESS accurate than ultrasound dating, this does not mean it cannot be useful, in particular for promoting the very much needed interventions this review is discussing. I am aware the WHO in their ANC guidelines recommend early ultrasound assessment for all women, and the current recommendation for interventions for preterm birth likely have to refer to these, However, for this article I find this problematic. And you might perhaps see this as a form of bias. At the bear minimum the recommendation and therefore this article needs to offer LMIC that do not have ability to provide all women with an early scan an alternative. Table 1 stating that WHEN ACS can be provided is only when GA is accurately assessed through ultrasound, makes the rest of the text and review completely inaccessible and useless for countries that won't be able to live up to this recommendation. And this means many women and newborns that could potentially benefit from the interventions in this review, will miss these opportunities. 

A reflection on the barriers of global guidelines - that are very much based on evidence generated in high income settings and how these are not always transferable to LMIC settings could strengthen the review, in particular as the authors emphasise the importance of implementation science. Stressing the importance of the need for contex-adapting guidelines - and thereby perhaps creating a platform for discussing optimal treatment - is recommended. Perhaps in some countries this means focussing on those women that will have an expected pre-term birth (women with Pre eclampsia for example). This also means guidelines need to be integrated or connected in some way. There are major research gaps in this field in LMIC setting and the authors could stress the importance of conducting studies in these settings.

Reviewer #3: Thank you for the opportunity to review an important and high-quality piece of work, thus I have only minor comments/suggestions for publication. Authors conducted a complex mixed methods review and narrative synthesis and assessed contextual factors and implementation strategies influencing use of antenatal corticosteroids, tocolytics, magnesium sulphate, antibiotics, for management of women at risk of preterm birth.

1) This is an important implementation research question in the maternal public/global health and preterm birth field and authors made it very clear in the background section. Perhaps authors could add in the background the rational for selecting those 4 (just clinical) interventions that are used during pregnancy or just before preterm birth (2015 WHO guideline on recommendations on interventions to improve preterm birth outcomes is only mentioned in methods). Also, a more recent Cochrane review also found other clinical interventions (e.g. screening for lower genital tract infections; zinc supplementation, cervical stitch for women at high risk of preterm birth) had a clear benefit on preterm birth outcomes (Medley et al 2018 - Interventions during pregnancy to prevent preterm birth: an overview of Cochrane systematic reviews). I think the four interventions selected by authors will be targeting a specific population group, pregnant women at risk of preterm birth (who may or may not necessarily experience a preterm birth at the end).

2) The methods section is very comprehensive, and authors report the review in accordance with recommended guidelines. There are not important deviations from the prospero registration. It is innovative to see how qualitative and quantitative data are mapped together and how identified contextual factors are then mapped to TDF and COM-B models of behaviour change to identify strategies - authors could explain briefly why they selected those two models and not others and how/why those two behaviours (appropriate use by providers, acceptability by women) were chosen for the purposes of the mapping?

3) Findings are very clearly presented in the text ,tables. Since most of the included studies in the review are from HIC, would it be useful for readers to know where the data from the overarching themes and strategies came from and understand variations from different country income levels? (e.g. in the 'Limitations about determining gestational age' - authors do mention data from LMIC but not in other themes?). Just a suggestion but I understand it may not be possible for pragmatic purposes! And this is part of the discussion.

4) Figure 5: Mapping factors affecting health providers' appropriate use of interventions for preterm birth management using COM-B - Not sure why implementation strategies are inside the facilitator boxes (2nd column)? Would it be more appropriate to add a third column with implementation strategies (barriers/facilitators/implementation strategies) - you would expect both barriers and facilitators to inform implementation strategies? Eg confusion about different courses, doses etc can be a barrier and can inform eg the development of education and training for care providers (to address that barrier).

5) Discussion, limitations and implications for practice and research are very well presented. Minor comment on wording: the review sometimes uses 'preterm

birth interventions' or uses preterm birth indistinctively with preterm labour and this may need to be revised e.g. a woman may go into preterm labour and have some interventions for management (thus at risk of preterm birth) but may not end up experiencing a preterm birth. I am not sure how appropriate is to use 'threatened preterm birth' (but threatened preterm labour) or 'preterm birth interventions' (but eg. interventions during pregnancy to reduce preterm birth)

6) It is clear this study findings will be helpful for policy makers and implementers who are planning policy to implement and scale up clinical interventions to reduce preterm birth (I think important abstract findings only mention barriers maybe due to word count? I think facilitators and strategies are also very mportant). Congratulations to authors for such a great work!

[LINK]

---

## [Decision Letter · Decision Letter 2]

27 Jun 2022

Dear Dr. Bohren,

Thank you very much for re-submitting your manuscript "Factors influencing appropriate use of interventions for management of women experiencing preterm birth: a mixed-methods systematic review and narrative synthesis" (PMEDICINE-D-21-05120R2) for review by PLOS Medicine.

I have discussed the paper with my colleagues and the academic editor and it was also seen again by two reviewers. I am pleased to say that provided the remaining editorial and production issues are dealt with we are planning to accept the paper for publication in the journal.

[LINK]

We look forward to receiving the revised manuscript by Jul 04 2022 11:59PM.   

Sincerely,

Beryne Odeny, PhD

PLOS Medicine

plosmedicine.org

Requests from Editors:

1) Abstract - please trim down the background section to 3-4 sentences.

2) Author summary – please trim the 3rd bullet under “What did the researchers do and find?” This should contain at most 4 sentences.

3) Please use a uniform black color for main text, headings, and subheadings, table (lines and text)

4) PRISMA and ENTREQ checklists- when completing the checklists, please use both section and paragraph numbers, rather than section headers only.

5) Please remove the ‘Sources of funding,” “Competing interest” statements from the end of the main text. In the event of publication, this information will be published as metadata based on your responses to the submission form.

Comments from Reviewers:

Reviewer #1: I think the authors have answered my questions and have addressed concerns of mine and of other reviewers adequately. Once the manuscript has met all the editorial requirements, I think it can be accepted. Well done and congratulations to the authors.

Reviewer #3: Thank you again for the opportunity to review this revision. I think authors have addressed well most comments and the manuscript has significantly improved. I would suggest some minor changes before accepting for publication. 

Abstract and author summary: the paper is assessing assess factors (barriers and facilitators) affecting the appropriate use of 4 key interventions to prevent or manage preterm birth and identifying potential strategies - why findings in these two sections present only barriers? Suggest briefly including findings on all, barriers, facilitators and potential strategies. 

I am still trying to understand Figure 4 and 6 and I think a small revision will be helpful here - authors report they map factors (barriers, facilitators/mixed evidence) affecting health providers' appropriate use and women's acceptance of interventions for preterm birth management using behavioural frameworks. However, figures can be misleading by reporting 'implementation strategies' within the 'facilitators' boxes. I presume those 'facilitators' and 'implementation strategies' were from the perspectives of the participants from the included studies (it is surprising to see that no included study suggested implementation strategies for specific barriers?). However, in the field of implementation research, there is a clear distinction between 'barriers and facilitators' (aka 'determinants of implementation or any contextual factors external to an intervention that affect implementation success) and 'implementation strategies' (methods/techniques used to enhance the adoption, implementation, and sustainability of an intervention). Thus, barriers & facilitators/ contextual factors inform the appropriate selection of implementation strategies. 

I understand other reviewers suggested to shorten the discussion section and it is now more comprehensive and less repetitive. But not sure about moving the 'implications for practice and research' to an appendix, this is a very important section of any discussion and a short summative paragraph in the main paper will be extremely valuable. Good luck!

[LINK]

---

## [Editor Report · Decision Letter 3]

12 Jul 2022

Dear Dr Bohren, 

On behalf of my colleagues and the Academic Editor, Dr. Sarah J Stock, I am pleased to inform you that we have agreed to publish your manuscript "Factors influencing appropriate use of interventions for management of women experiencing preterm birth: a mixed-methods systematic review and narrative synthesis" (PMEDICINE-D-21-05120R3) in PLOS Medicine.

PRESS

Sincerely, 

Beryne Odeny 

PLOS Medicine